# DiffInfinite: Large Mask-Image Synthesis via Parallel Random Patch Diffusion in Histopathology

**Marco Aversa**
University of Glasgow and Dotphoton
Glasgow, United Kingdom
marco.aversa@glasgow.ac.uk

**Gabriel Nobis**
Fraunhofer HHI
Berlin, Germany
gabriel.nobis@hhi.fraunhofer.de

**Miriam Hägele**
Aignostics
Berlin, Germany
miriam.haegele@aignostics.com

**Kai Standvoss**
Aignostics
Berlin, Germany
kai.standvoss@aignostics.com

**Mihaela Chirica**
Institute of Pathology, LMU Munich
Munich, Germany
mihaela.chirica@med.uni-muenchen.de

**Roderick Murray-Smith**
University of Glasgow
Glasgow, United Kingdom
roderick.murray-smith@glasgow.ac.uk

**Ahmed Alaa**
UC Berkeley
Berkeley, California
amalaa@berkeley.edu

**Lukas Ruff**
Aignostics
Berlin, Germany
lukas.ruff@aignostics.com

**Daniela Ivanova**
University of Glasgow
Glasgow, United Kingdom
daniela.ivanova@glasgow.ac.uk

**Wojciech Samek**
Fraunhofer HHI and TU Berlin
Berlin, Germany
wojciech.samek@hhi.fraunhofer.de

**Frederick Klauschen**
Institute of Pathology, LMU Munich
Munich, Germany
f.klauschen@lmu.de

**Bruno Sanguinetti**
Dotphoton
Zug, Switzerland
bruno.sanguinetti@dotphoton.com

**Luis Oala**
Dotphoton
Zug, Switzerland
luis.oala@dotphoton.com

## Abstract

We present DiffInfinite, a hierarchical diffusion model that generates arbitrarily large histological images while preserving long-range correlation structural information. Our approach first generates synthetic segmentation masks, subsequently used as conditions for the high-fidelity generative diffusion process. The proposed sampling method can be scaled up to any desired image size while only requiring

37th Conference on Neural Information Processing Systems (NeurIPS 2023) Track on Datasets and Benchmarks.

small patches for fast training. Moreover, it can be parallelized more efficiently than previous large-content generation methods while avoiding tiling artifacts. The training leverages classifier-free guidance to augment a small, sparsely annotated dataset with unlabelled data. Our method alleviates unique challenges in histopathological imaging practice: large-scale information, costly manual annotation, and protective data handling. The biological plausibility of DiffInfinite data is evaluated in a survey by ten experienced pathologists as well as a downstream classification and segmentation task. Samples from the model score strongly on anti-copying metrics which is relevant for the protection of patient data.

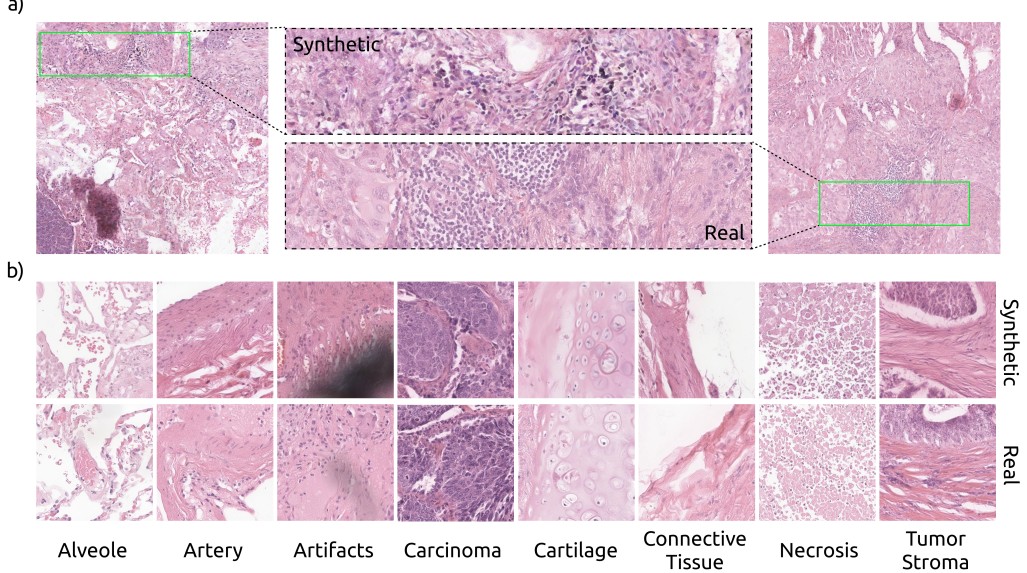

Figure 1: a) Examples of synthetic and real $2048 \times 2048$ images. b) Pairs of $512 \times 512$ synthetic tiles (top) with the closest real images found with Inception-v3 near-neighbour (bottom).

# 1 Introduction

Deep learning (DL) models are promising auxiliary tools for medical diagnosis [1–3]. Applications like segmentation and classification have been refined and pushed to the limit on natural images [4]. However, these models trained on rich datasets still have limited applications in medical data. While segmentation models rely on sharp object contours when applied to natural data, in medical imaging, the model struggles to detect a specific feature because it has a "limited ability to handle objects with missed boundaries" and often "miss tiny and low-contrast objects" [5, 6]. Therefore, task-specific medical applications require their own specialised and fine-grained annotation. Data labelling is arguably one of the most critical bottlenecks in healthcare machine learning (ML) applications. In histopathology, pathologists examine the histological slide at multiple levels, usually starting with a lower magnification to analyse the tissue architecture and cellular arrangement and gradually proceeding to a higher magnification to examine cell morphology and subcellular features, such as the appearance and number of nucleoli, chromatin density and cytoplasm appearance. Annotating features within gigapixel whole slide images (WSIs) with this level of detail demands effort and time, often leading to sparse, limited annotated data. In addition, due to privacy regulations and ethics [7, 8], having access to medical data can be challenging since it has been shown that it is possible to extract patients' sensitive information [9] from this data.

In histopathology, state-of-the-art ML models require the context of the entire WSIs, with features at different scales, in order to distinguish between different tumor sub-types, grades and stages [10]. Despite the demonstrated effectiveness of diffusion models (DMs) in generating natural images compared to other approaches, they still have rarely been applied in medical imaging. Existing

generative models in histopathology can generate images of relatively small resolution compared to WSIs. To give a few examples, the application of Generative Adversarial Networks (GANs) in cervical dysplasia detection [11], glioma classification [12], and generating images of breast and colorectal cancer [13], generate images with $256 \times 128$ px, $384 \times 384$ px and $224 \times 224$ px, respectively. In spite of their current limitations in generating images at scales necessary to fully address all medical concerns, the use of synthetic data in medical imaging can provide a valuable solution to the persistent issue of data scarcity [14–17]. Models generally improve after data augmentation and synthetic images are equally informative as real images when added to the training set [18, 19]. Data augmentation could also help with the underrepresentation in data sets of rare cancer subtypes. By adding synthetic images to the training set, Chen et al. [20] demonstrated that their model had better accuracy in detecting chromophobe renal cell carcinoma, which is a rare subtype of renal cell carcinoma. Furthermore, Doleful et al. [21] showed how synthetic histological images could be used for educational purposes for pathology residents. Regarding the challenges highlighted before, we present a novel sampling method to generate large histological images with long-range pixel correlation (see Fig. 1), aiming to extend up to the resolution of the WSI.

Our contributions are as follows: 1) We introduce DiffInfinite, a hierarchical generative framework that generates arbitrarily large images, paired with their segmentation masks. 2) We introduce a fast outpainting method that can be efficiently parallelized. 3) The quality of DiffInfinite data is evaluated by ten experienced pathologists as well as downstream machine learnings tasks (classification and segmentation) and anti-duplication metrics to assess the leakage of patient data from the training set.

## 2 Related Work

Large-content image generation can be reduced to inpainting/outpainting tasks. Image inpainting is the problem of reconstructing unknown or unwanted areas within an image. A closely related task is image outpainting, which aims to predict visual content beyond the boundaries of an image. In both cases, the newly in- or outpainted image regions have to be visually indistinguishable with respect to the rest of the image. Such image completion approaches can help utilise models trained on smaller patches for the purpose of generating large images, by initially generating the first patch, followed by its extension outward in the desired direction.

**Traditional approaches**    Traditional methods for image region completion rely on repurposing known image features, necessitating costly nearest neighbour searches for suitable pixels or patches [22–26]. Such methods often falter with complex or large regions [24]. In contrast, DL enables novel, realistic image synthesis for inpainting and outpainting. Some methods like Deep Image Prior [27] condition new image areas on the existing image, while others aim to learn natural image priors for realistic generation [28, 29].

**Generative modelling for conditional image synthesis**    GANs have dominated image-to-image translation tasks like inpainting and outpainting for years [28–42]. Recently, DMs have surpassed GANs in various image generation tasks [43]. Palette [44] was the first to apply DMs to tasks like inpainting and outpainting. RePaint [45] and ControlNet [46] demonstrate resampling and masking techniques for conditioning using a pre-trained diffusion model. SinDiffusion [47] and DiffCollage [48] offer state-of-the-art outpainting solutions using DMs trained with overlapping patches. In parallel to our work, Bond-Taylor and Willcocks [49] developed a related approach called $\infty$-Diff which trains on random coordinates, allowing the generation of infinite-resolution images during sampling. However, in contrast to our approach the method does not involve image compression in a latent space.

**Synthetic data assessment**    The authenticity of synthetic data produced by DMs, trained on vast paired labelled datasets [50], remains contentious. Ethical implications necessitate distinguishing if generated images are replicas of training data [51, 52]. The task is complicated due to subjective visual similarities and diverse dataset ambiguities. Various metrics have been proposed for quantifying data replication, including information theory distances from real data [53], consistency measurements using downstream models [54, 55], comparison with inpainted areas [52], and detection of "forgotten" examples [56].

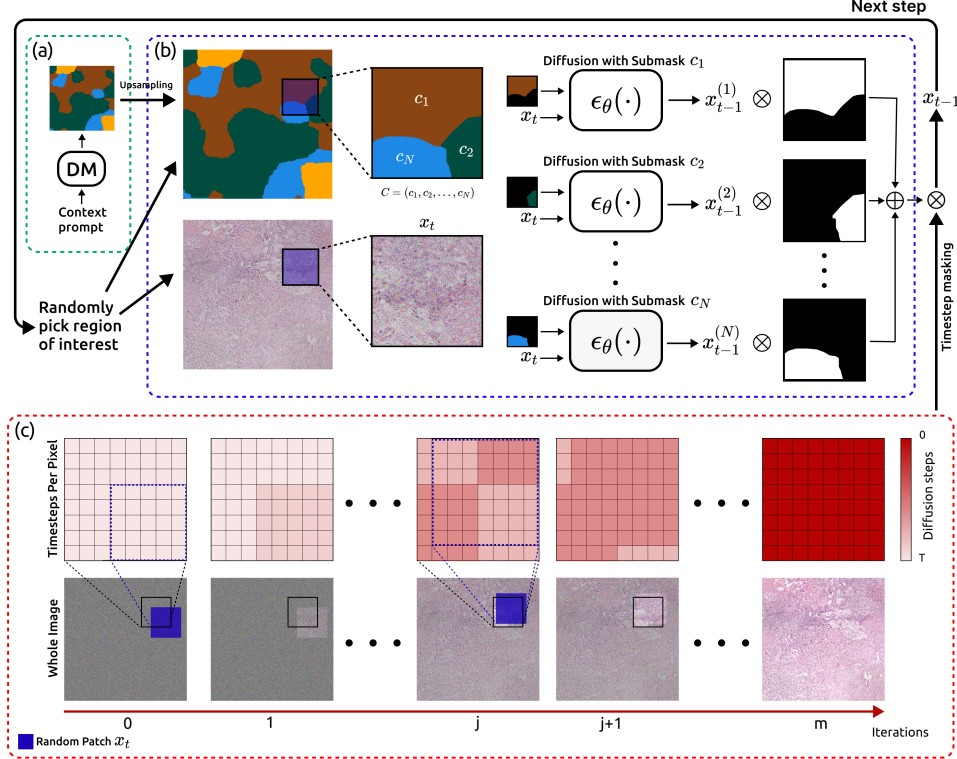

Figure 2: DiffInfinite generation method. a) Large-scale context mask generation. A diffusion model conditioned on a large-scale conditional prompt (e.g. Adenocarcinoma subtype) generates a low-resolution mask. The mask is upsampled via linear interpolation to the desired image size. b) Diffusion steps on large images. Given a random position, we select a sub-tile with its segmentation mask. A diffusion model generates in parallel the next step conditioned on each conditional label, or prompt, found in the mask. The outputs are masked individually with the corresponding label. The next step is the union of all the sub-patches. c) Tracking time steps pixel-wise. We keep track of the time step of each pixel in the large image. The model evolves only the pixels with the higher time step on each iteration.

## 3 Preliminaries

**Diffusion Models** DMs [57–59] represent a class of parameterized Markov chains that effectively optimize the lower variational bound associated with the likelihood function of the unknown data distribution. By iteratively adding small amounts of noise until the image signal is destroyed and then learning to reverse this process, DMs can approximate complex distributions much more faithfully than GANs [60]. The increased diversity of samples while preserving sample fidelity comes at the cost of training and sampling speed, with DMs being much slower than GANs [43]. The universally adopted solution to this problem is to encode the images from pixel space into a lower dimensional latent space via a Vector Quantised-Variational AutoEncoder (VQ-VAE), and perform the diffusion process over the latents, before decoding back to pixel space [61]. Pairing this with the Denoising Diffusion Implicit Models (DDIMs) sampling method [62] leads to faster sampling while preserving the DM objective

$$z_{t-1} = \sqrt{\alpha_{t-1}} \left( \frac{z_t - \sqrt{1-\alpha_t}\epsilon_\theta(z_t,t)}{\sqrt{\alpha_t}} \right) + \sqrt{1 - \alpha_{t-1} - \sigma_t^2}\epsilon_\theta(z_t,t) + \sigma_t\epsilon_t, \qquad (1)$$

where $z_t$ is the latent variable at time step $t$ in the VQ-VAE latent space, $\alpha_t$ is the noise scheduler, $\epsilon_\theta$ is the noise learned by the model and $\epsilon_t$ is random noise. Conditioning can be achieved either by specifically feeding the condition with the noised data [44, 63], by guiding an unconditional model using an external classifier [64, 65] or by classifier-free guidance [66] used in this work, where the

convex combination

$$\tilde{\epsilon}_\theta(z_t, c) = (1 + \omega)\epsilon_\theta(z_t, c) - \omega\epsilon_\theta(z_t, \varnothing), \tag{2}$$

of a conditional diffusion model $\epsilon_\theta(z_t, c)$ and an unconditional model $\epsilon_\theta(z_t, \varnothing)$ is used for noise estimation. The parameter $\omega$ controls the tradeoff between conditioning and diversity, since $\omega > 0$ introduces more diversity in the generated data by considering the unconditional model while $\omega = 0$ uses only the conditional model.

# 4 Infinite Diffusion

The DiffInfinite approach we present here[1], is a generative algorithm to generate arbitrarily large images without imposing conditional independence, allowing for long-range correlation structural information. The method overcomes this limitation of DMs for large-content generation by deploying multiple realizations of a DM on smaller patches. In this section, we first define a mathematical description of this hierarchical generation model and then describe the sampling method paired with a masked conditioned generation process.

## 4.1 The Method

Let $X \sim \mathcal{X}$ be a large-content generating random variable taking values in $\mathbb{R}^{KD}$. Using the approach of latent diffusion models [61], the high-dimensional content is first mapped to the latent space $\mathbb{R}^D$ by $\Phi(X) = Y \sim \mathcal{Y}_\Phi$. For simplicity, we assume throughout this work the existence of an ideal encoder-decoder pair $(\Phi, \Psi)$ such that $\Psi(\Phi(X)) = X$ is the identity on $\mathbb{R}^{KD}$. Assume further, to have a reverse time model $(SM_\theta, \epsilon_\theta)$ at hand consisting of a sampling method $SM_\theta$ and a learned model $\epsilon_\theta$ trained on small patches $Z \sim \mathcal{Z}_\Phi$ taking values in $\mathbb{R}^d$. The reverse time model transforms $z_T \sim \mathcal{N}(0, I_d)$ over the time steps $t \in \{T, T-1, ..., 1\}$ recursively by

$$z_{t-1} = SM_\theta(z_t) \tag{3}$$

to an approximate instance of $\mathcal{Z}_\Phi$. We aim to sample instances from $\mathcal{Y}_\Phi$ by deploying multiple realizations of the reverse time model $(SM_\theta, \epsilon_\theta)$. Towards that goal, define the set of projections

$$\mathcal{C} := \{proj_I : \mathbb{R}^D \to \mathbb{R}^d \mid I \subset \mathbb{N} \text{ correspond to } d \text{ indices of connected pixels in } \mathbb{R}^D\}, \tag{4}$$

where $proj \in \mathcal{C}$ models a crop $proj(Y) \in \mathbb{R}^d$ of $d$ connected pixels from the latent image $Y$. Since the model $\epsilon_\theta$ is trained on images taking values in $\mathbb{R}^d$ the standing assumption is

**Assumption 1** *Any projection $proj \in \mathcal{C}$ maps $Y$ to the same distribution $proj(Y) \sim \mathcal{Z}_\Phi$ in $\mathbb{R}^d$.*

Since the goal is to approximate an instance of $\mathcal{Y}_\Phi$, we initialize the sampling method by $y_T \sim \mathcal{N}(0, I_D)$ and proceed in the following way: Given $y_t$, randomly choose $proj_{I_1}, ..., proj_{I_m} \in \mathcal{C}$ independent of the state $y_t$ such that $proj_{I_1}, ..., proj_{I_m}$ are non equal crops that cover all latent pixels in $\mathbb{R}^D$. To be more precise, for every $i \in \{1, ..., D\}$ we find at least one $j \in \{1, ..., m\}$ with $i \in I_j$. For every projection $proj_{I_1}, ..., proj_{I_m}$ we calculate the crop $z_t^j = proj_{I_j}(y_t)$ of the current state $y_t$ and perform one step of the reverse time model following the sampling scheme

$$z_{t-1}^j = SM_\theta(z_t^j), \quad j \in \{1, ..., m\}. \tag{5}$$

This results in overlapping estimates $z_{t-1}^1, ..., z_{t-1}^m$ of the subsequent state $t-1$ and we simply assign to every pixel in the latent space the first value computed for this pixel such that

$$[y_{t-1}]_i = [z_{t-1}^j]_l, \quad \text{where } j = \min\{j' \mid i \in I_{j'}\} \tag{6}$$

and $l$ refers to the entry in $z_{t-1}^j$ corresponding to $i$ with $[proj_{I_j}(y_{t-1})]_l = [y_{t-1}]_i$. Hence, starting from $y_T \sim \mathcal{N}(0, I_D)$ we sample in the first step from a distribution

$$y_{T-1} \sim p_{T-1,\theta}(y \mid y_T, proj_{I_1}, ..., proj_{I_m}). \tag{7}$$

Using Bayes' theorem, this distribution simplifies to

$$p_{T-1,\theta}(y \mid y_T, proj_{I_1}, ..., proj_{I_m}) = p_{T-1,\theta}(y \mid y_T), \tag{8}$$

---

[1]Code available at https://github.com/marcoaversa/diffinfinite

since we sample the projections independently from $y_T$. Repeating the argument, we sample in every step from a distribution $y_{t-1} \sim p_{t-1,\theta}(y|y_t, ..., y_T)$ over $\mathbb{R}^D$ instead of sampling from $z_{t-1} \sim q_{t-1,\theta}(z|z_t, ..., z_T)$ over $\mathbb{R}^d$. Hence, we approximate the true latent distribution $\mathcal{Y}_\Phi$ by the approximate distribution with density $p_{0,\theta}(y|y_1, ..., y_T)$. In contrast to [48], our method does not use the assumption of conditional independence and the method can be applied to a wide range of DMs, without an adjustment of the training method. As the authors of [48] point out in their section on limitations, the assumption of conditional independence is not well-suited in cases of a data distribution with long-range dependence. For image generation in the medical context, we aim to circumvent this assumption as we do not want to claim that the density of a given region depends only on one neighboring region. The drawback of dropping the assumption is that we only approximate the reverse time model of the latent image distribution $\mathcal{Y}_\Phi$ indirectly, by multiple realizations of a reverse time model that approximates $\mathcal{Z}_\Phi$.

## 4.2 Semi-supervised Guidance

In order to generate diverse high-fidelity data, DMs require lots of training data. Perhaps, training on a few samples still extracts significant features but it lacks variability, resulting in simple replicas. Here, we show how to enhance synthetic data diversity using classifier-free guidance as a semi-supervised learning method. In the classifier-free guidance [66], a single model is trained conditionally and unconditionally on the same dataset. We adapt the training scheme using two separate datasets. The model is guided by a small and sparsely annotated dataset $q_1$, used for the conditional training step, while extracts features by the large unlabelled dataset $q_0$, used on the unconditional training step (see Alg.1)

$$(z_0, c) = \begin{cases} (z_0, \varnothing) \sim q_0(z_0) & \text{if } u \geqslant p_{unc} \\ (z_0, c) \sim q_1(z_0, c) & \text{otherwise} \end{cases}, \tag{9}$$

where $u$ is sampled from a uniform distribution in [0,1], $p_{unc}$ is the probability of switching from the conditional to the unconditional setting and $\varnothing$ is a null label. During the sampling, a tradeoff between conditioning and diversity is controlled via the parameter $\omega$ in eq.2.

## 4.3 Sampling

**High-level content generation**  The outputs of DMs have pixel consistency within the training image size. Outpainting an area with a generative model might lead to unrealistic and odd artifacts due to poor long-range spatial correlations. Here, we show how to predict pixels beyond the image's boundaries by generating a hierarchical mapping of the data. The starting point is the generation of the highest-level representation of the data. In our case, it is the sketch of the cellular arrangement in the WSI (see Figure 2a). Since higher-frequency details are unnecessary at this stage, we can downsample the masks until the clustering pattern is still recognizable. The diffusion model, conditioned on the context prompt (e.g. Adenocarcinoma subtype), learns the segmentation masks which contain the cellular macro-structures information.

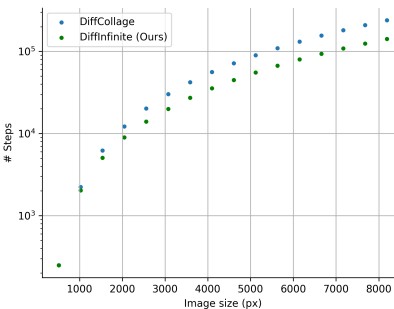

Figure 3: Comparison of sampling speed for DiffCollage and DiffInfinite, measuring diffusion steps required for image sampling. Demonstrating increased efficiency of DiffInfinite for larger images.

**Random patch diffusion**  Given a segmentation mask $M$, we can proceed with the large image sampling according to Section 4.1 in the latent space $\mathbb{R}^D$ of $Y = \Phi(X)$ (see Alg.2). Since we trained a conditional diffusion model with conditions $c_1, ..., c_N$, the learned model takes the form $\epsilon_\theta(z_t, t) = (\epsilon_\theta(z_t, t|c_1), ..., \epsilon_\theta(z_t, t|c_N))$. Given $y_t$, we first sample projections $proj_{I_1}, ..., proj_{I_m} \in \mathcal{C}$, corresponding to different crops of $d$ connected pixels up to the $m$-th projection with $\cup_{j=1}^m I_j = \{1, ..., D\}$ and $\cup_{j=1}^m I_j \setminus \cup_{j=1}^{m-1} I_j \neq \varnothing$ (see the left hand-side of Figure 2b). Note that $m$ is not fixed, but varies over the sampling steps and is upper bounded by the number of possible crops of $d$ connected pixels. The random selection of

the projection is implemented such that regions with latent pixels of low projection coverage are more likely. Secondly, we calculate for every projection $j \in \{1, ..., m\}$ the crop $proj_{I_j}(y_t)$ and perform one step of the DDIM sampling procedure using the classifier-free guidance model $(1 + \omega)\epsilon_\theta(z_t^j, t, c) - \omega\epsilon_\theta(z_t^j, t, \varnothing)$, where $\epsilon_\theta$ is the learned model and $z_t^j = proj_{I_j}(y_t)$. This results for every pixels $i \in I_j$ in $N$ values $DDIM_{\theta, c_1}(proj_{I_j}(y_t)), ..., DDIM_{\theta, c_N}(proj_{I_j}(y_t))$, one for every condition $c_i$ (see the right hand-side of Figure 2b). If $i \notin I_{j'}$ for all $j'$, the pixel $i$ has not been considered yet and we assign $i$ the value $[y_{t-1}]_i = [DDIM_{\theta, M_i}(proj_{I_j}(y_t))]_l$, where $l$ corresponds to the pixel $i$ under the projection $I_j$ and $M_i$ is the value of $i$ in the mask $M$. Since we are updating random projections of the overall image, in the $t$-th step pixels either have the time index $t$ or $t + 1$, resulting in a reversed diffusion process of differing time states. We initialize a tensor $L_t$, with the same size $D$ as the latent variable, to keep track of the time index for each pixel. Each element is set to $L_T \equiv T$. In the $j$-th iteration of the $t$-th step we only update the pixels that have not been considered in one of the previous iterations of the $t$-th diffusion step, hence all the pixels in $i \in I_j$ with $proj_{I_j}(L_t)_i = t + 1$, similarly to the inpainting mask in the Repaint sampling method [45].

To restore the pixels that already received an update, i.e. every pixel $i \in I_j$ with $proj_{I_j}(L_t)_i = t$, we store a replica of the previous diffusion step for every pixel. Finally, we update all the time states in $L_t$ that received an update in the $j$-th iteration to $t$ resulting in $proj_{I_j}(L_t)_i = t$ for all $i \in I_j$. See the top row of Fig.2c for an illustration of the evolution of $L_t$. The random patch diffusion can also be applied to mask generation, where the only condition is the context prompt. This method can generate segmentation masks of arbitrary sizes with the correlation length bounded by two times the training mask image size.

**Parallelization** The sampling method proposed has several advantages. In Zhang et al. [48] each sequential patch is outpainted from the previous one with $50\%$ of the pixels shared. Here, the randomization eventually leads to every possible overlap with the neighboring patches. This introduces a longer pixel correlation across the whole generated image, avoiding artifacts due to tiling. In Figure 3, we show that the number of steps in the whole large image generation process is drastically reduced with the random patching method with respect to the sliding window one. Moreover, in the sliding sampling method, the model can be paralleled only 2 or 4 times, depending if we are outpainting the image horizontally or on both axis. In our approach, we can parallelize the sampling up to the computational resource limit.

---

**Algorithm 1** DiffInfinite Training

**Repeat**
1: Randomly train on labelled or unlabelled data with probability $p_{unc}$,
   $u \sim Uniform[0, 1]$
   $(z_0, c) = \begin{cases} (z_0, \varnothing) \sim q_0(z_0) & \text{if } u \geqslant p_{unc} \\ (z_0, c) \sim q_1(z_0, c) & \text{otherwise} \end{cases}$
2: Sample random time step
   $t \sim Uniform\{1, ..., T\}$
3: Sample noise, $\epsilon \sim \mathcal{N}(0, \mathbf{I})$
4: Corrupt data, $z_t = \gamma_t x_0 + \sigma_t \epsilon$
5: Take gradient descent step:
   $\nabla_\theta \|\epsilon - \epsilon_\theta(z_t, t, c)\|^2$
6: **until** converged

---

**Algorithm 2** DiffInfinite Sampling

**Input:** High-level segmentation mask $M \in \mathbb{R}^D$ and learned model $\epsilon_\theta$
**Output:** Synthetic image $X$ with the mask size
**Initialization:**
   $\mathbf{y}_T \sim \mathcal{N}(0, \mathbf{I})$, index set $I_0 = \varnothing$ and time state tensor $L_T \equiv T$
**Repeat**
1: **for** $t \in \{T - 1, ...0\}$ **do**
2:   **while** $\cup_{j=0}^m I_j \neq \{1, ..., D\}$ **do**
3:     $m \leftarrow m + 1$
4:     Select randomly $proj_{I_m} \in \mathcal{C} \backslash \{proj_{I_1}, ..., proj_{I_{m-1}}\}$
5:     Crop $z_t^m = proj_{I_m}(y_t)$
6:     **for** all conditions $n \in \{1, ..., N\}$ **do**
7:       DDIM sampling with classifier-free guidance
8:       $z_{t-1}^m | c_n \sim p_{\theta, t}(z | z_t^m, c_n)$
9:     **end for**
10:     **for** all indices $i \in I_m$ **do**
11:       **if** $i \notin I_j$ for all $j < m$ **then**
12:         $[y_{t-1}]_i \leftarrow [z_{t-1}^m | M_i]_l$
13:         $proj_{I_m}(L_t)_i \leftarrow t$
14:         such that $[proj_{I_m}(y_{t-1})]_l = [z_{t-1}^m | M_i]_l$
15:       **end if**
16:     **end for**
17:   **end while**
18: **end for**
19: $X \leftarrow \Psi(y_0)$

---

## 5 Data Assessment

To assess synthetic images for medical image analysis, we need to take various dimensions of data assessment into account. We extend traditional metrics from the natural image community with qualitative and quantitative assessments specific to the medical context. For the qualitative analysis, a team of pathologists evaluated the images for histological plausibility. The quantitative assessment entailed a proof-of-concept that a model can learn sensible features from the synthetically generated image patches for a relevant downstream task. As data protection is highly relevant regarding patient data, we performed evaluations to rule out memorization effects of the generative model.

Table 1: Metrics to quantitatively evaluate the quality of the generated images. Left: scores for images of size $512 \times 512$. DiffInfinite (a) first generates a mask and secondly an image following Section 4.1. Right: scores for real and generated images of size $2048 \times 2048$ resized to $512 \times 512$. All methods use the same model trained on small patches of size $512 \times 512$. DiffCollage corresponds to the method proposed in [48]. DiffInfinite (b) uses the real masks, while DiffInfinite (c) first generates a mask and secondly the large image. DiffInfinite (b) & (c) refers to the mixture of the generated dataset from DiffInfinite (b) and DiffInfinite (c).

|  | IP ↑ | IR ↑ | IS ↑ | FID ↓ |
|---|---|---|---|---|
| Morph-Diffusion [72] | 0.26 | **0.85** | 2.1 | 20.1 |
| NASDM [73] | - | - | **2.7** | **15.7** |
| DiffInfinite (a) | **0.94** | 0.70 | **2.7** | 26.7 |

|  | IP ↑ | IR ↑ |
|---|---|---|
| DiffCollage | 0.94 | 0.22 |
| DiffInfinite (b) | 0.95 | **0.48** |
| DiffInfinite (c) | **0.98** | 0.44 |
| DiffInfinite (b) & (c) | **0.98** | 0.33 |

## 5.1 Traditional Fidelity

We evaluate the fidelity of synthetic $512 \times 512$ images by calculating Improved Precision (IP) and Improved Recall (IR) metrics between 10240 real and synthetic images [67].[2] The IP evaluates synthetic data quality, while the IR measures data coverage. Despite their unsuitability for histological data [68, 69], Frechet-Inception Distance (FID) and Inception Score (IS) [70, 71] are reported for comparison with [72] and Shrivastava and Fletcher [73].[3] The metrics' explanations and formulas can be found in Appendix C.

In Table 1 (left), we report an IP of $0.94$ and an IR of $0.70$, indicating good quality and coverage of the generated samples. However, we note that these metrics are only somewhat comparable due to the different types of images generated by MorphDiffusion [72] and NASDM [73]. For the large images of size $2048 \times 2048$, we rely solely on the IP and IR for quantitative evaluation due to the limited number of 200 generated large images. As shown in Figure 3(a) of [67], FID is unsuitable for evaluating such a small sample size, while IP and IR are more reliable. In Table 1 (right), we find that generating images first results in slightly higher IR, while generating the mask first achieves an IP of $0.98$. For the sake of completeness we also report the scores then combining the two datasets. To compare our method to DiffCollage we generate 200 images using [48]. DiffInfinite performs better than DiffCollage wrt. to IP and IR. The drop of IR to $0.22$ might be a result of the tiling artifacts observable in the LHS of Figure 11.

## 5.2 Domain Experts Assessment

To assess the histological plausibility of our generated images, we conducted a survey with a cohort of ten experienced pathologists, averaging 8.7 years of professional tenure. The pathologists were tasked with differentiating between our synthetized images and real image patches extracted from whole slide images. We included both small patches (512 × 512 px) commonly used for downstream tasks as well as large patches (2048 × 2048 px). Including large patches enabled us to additionally evaluate the modelled long-range correlations in terms of transitions between tissue types as well as growth patterns which are usually not observable on the smaller patch sizes but essential in histopathology. In total the survey contained 60 images, in equal parts synthetic and real images as well as small and large patches. The overall ability of pathologists to discern between real and synthetic images was modest, with an accuracy of 63%, and an average reported confidence level of 2.22 on a 1-7 Likert scale. While we observed high inter-rater variance, there was no clear correlation between experience and accuracy (r(8) = .069, p=.850), nor between confidence level and accuracy (r(8) = .446, p=.197). Furthermore there was no significant correlation between the participants' completion time of the survey and the number of correct responses (r(8) = -.08, p=.826).

Surprisingly, we found a similar performance for both, real and synthetic images. This indicates that, while clinical practice is mostly based on visual assessment, it is not a common task for pathologists to be restricted to parts of the whole slide image only. More detailed visualizations of the individual scores can be found in Appendix B. Besides this satisfactory result, we additionally wanted to

---

[2] https://github.com/blandocs/improved-precision-and-recall-metric-pytorch
[3] https://github.com/toshas/torch-fidelity

Table 2: Zero-shot evaluation results of the downstream tasks, encompassing both classification and segmentation scenarios. We employed three distinct models for each scenario: The first, "Trained Real," was trained using real data (in-house IH1), which also served as the training set for DiffInfinite. The second, "Trained Synthetic," was trained using samples generated from DiffInfinite, and the third, "Trained Augmented," utilized a combination of real and synthetic data. Our evaluation extends across separate lung cohorts (internal datasets IH2 and IH3) and additional indications (external datasets NCT, CRC, PCam), with varying degrees of data drift introduced.

| | IH1 | IH2 | IH3 | NCT-100K | CRC-7K | PCam-327K |
|---|---|---|---|---|---|---|
| Drift components | - | Patient change Different center | | Patient change Different center Indication change | | Patient change Different center Indication change Lower resolution |
| Trained Real | $0.846 \pm 0.005$ | $0.733 \pm 0.021$ | $0.598 \pm 0.049$ | $\mathbf{0.857} \pm 0.009$ | $\mathbf{0.822} \pm 0.034$ | $0.628 \pm 0.035$ |
| Trained Synthetic | $0.747 \pm 0.025$ | $\mathbf{0.753} \pm 0.005$ | $\mathbf{0.699} \pm 0.002$ | $0.796 \pm 0.023$ | $0.753 \pm 0.038$ | $0.628 \pm 0.012$ |
| Trained Augmented | $\mathbf{0.852} \pm 0.007$ | $0.732 \pm 0.027$ | $0.637 \pm 0.025$ | $0.847 \pm 0.044$ | $0.811 \pm 0.057$ | $\mathbf{0.641} \pm 0.035$ |

(a) Classification results

| | IH2 |
|---|---|
| Trained Real | $0.614 \pm 0.009$ |
| Trained Synthetic | $0.471 \pm 0.039$ |
| Trained Augmented | $\mathbf{0.710} \pm 0.021$ |

(b) Segmentation results

explore the limitations of our method by assessing the nuanced differences pathologists observed between synthetic and real images. While overall the structure and features seemed similar and hard to discern, they sometimes reported regions of inconsistent patterns, overly homogeneous chromatin in some of the synthetic nuclei, peculiarities in cellular and intercellular structures, and aesthetic elements. These seemed to be especially pronounced in tumorous regions where sometimes the tissue architecture appeared exaggerated, the transition to stroma or surrounding tissue was too abrupt and some cells lacked distinguishable nucleoli or cytoplasm. We attribute the nuanced effect of larger image size on the accuracy on this observation (cf. Fig. 5C). Overall the finding of the conducted survey demonstrates how complex the task of distinguishing between real and synthetically generated data is even for experienced pathologists while still highlighting potential areas to improve the generative model.

## 5.3 Synthetic Data for Downstream Tasks

A major interest in the availability of high quality labeled synthetic images is their use in downstream digital pathology applications. In this area, two primary challenges are the binary classification of images into cancerous or healthy tissues and the segmentation of distinct tissue areas in the tumor microenvironment. The unique ability of our technique to generate images of different cancer subtypes through the context prompt as well as the ability to create new segmentation masks and their corresponding H&E images specifically addresses these two challenges. Notably, expert annotations are costly and time consuming to acquire thus emphasizing the benefits of being able to train on purely synthetic datasets or augmenting annotated data in the low data regime. To showcase these two usecases we performed a series of experiments in both classification and segmentation settings. For all experiments, we trained a baseline classier on a relatively small number of expert annotations IH1 (#patches = 3726)) — the same that were used to train DiffInfinite — and additionally trained one model purely on synthetic data (IH1-S, #patches = 9974, $\omega = 0$), and one model on the real data augmented with the synthetic images. To generate target labels for the classification experiments, we simplified the segmentation challenge by categorizing patches with at least 0.05% of pixels labeled as 'Carcinoma' in the segmentation masks as 'Carcinoma'. All other patches were labeled 'Non-Carcinoma'. We evaluated all three classification models on several out-of-distribution datasets. We utilized two proprietary datasets (from the same cancer type with similar attributes but from distinct patient groups: IH1 (# patients=13, # patches=704) and IH3 (# patients=2, # patches=2817). Moreover, we assessed the models using two public datasets (NCK-CRC [74] and PatchCamelyon [75]), both representing tissue from different organs with distinct morphologies. Our findings, summarized in Table 2a, suggest that a classifier's out-of-distribution performance, trained with limited sample size and morphological diversity, can vary significantly (ranging from 0.628 to 0.857 balanced accuracy). This variability cannot be attributed solely to morphology but may also be influenced by

factors such as resolution and variations in scanning and staining techniques. Training exclusively with a larger set of synthetic images can enhance performance on some datasets (specifically IH2 and IH3), underscoring the advantages of leveraging the full training data in a semi-supervised manner within the generative model. Incorporating synthetic data as an augmentation to real data not only prevents the classifier's performance decline, as seen on NCT-CRC and Patchcamelyon, on similar datasets but also bolsters its efficiency on more distinct ones. For the more challenging segmentation task we again trained three segmentation models to differentiate between carcinoma, stroma, necrosis, and a miscellaneous class that included all other tissue types, such as artifacts. The baseline performance of the real data model on a distinct group of lung patients (dataset IH2) of a $F_1$ score of $0.614 \pm 0.009$ (across three random seeds) highlights the difficulty of generalizing out of distribution in this tasks. While the purely synthetic model was not able to fully recover the baseline performance ($0.471 \pm 0.039$), augmenting the small annotated dataset with synthetic data enhanced predictive performance to an $F_1$ score of $0.710 \pm 0.021$. This boost of 10 percentage points in performance demonstrates that the synthetic data provide new, relevant information to the downstream task. In summary, our findings demonstrate the feasibility of meeting or surpassing baseline performance levels for both tasks using either entirely synthetic data or within an augmented context. Nevertheless, the advantages of employing synthetic data in downstream tasks continue to pose a challenge, not only within the medical image domain but also across various other domains [76–78], thus requiring more comprehensive assessment and thorough examination.

### 5.4 Considerations on Memorization

In medicine the adherence to privacy regulations is a sensitive requirement. While it is generally not possible for domain experts to infer patient identities from the image content of a histological tile or slide alone [79], developers and users of generative models are well advised to understand the risk of correspondence between the training data and the synthesized data. To this end, we evaluate the training and synthesized data against two memorization measures. The authenticity score $A \in [0, 1]$ by [54] aims to measure the rate by which a model generates new samples (higher score means more innovative samples). Similarly, [80] aims to estimate the degree of data copying $C_T$ from the training data by the generative model. A $C_T \ll 0$ implies data copying, while a $C_T \gg 0$ implies an underfitting of the model. The closer to $0$ the better. See Appendix C for a precise closed form of the measures and Table 5 for the full quantitative results, indicating that the DiffInfinite model is not prone to data copying across all resolutions and variations considered here [4]. The $A$ range between $0.86$ and $0.98$, signifying a high rate of authenticity. While other papers unfortunately do not report such detailed memorization statistics for their models, the results by [54] suggest that a score $\gg 0.8$ is not trivial to achieve. None of the models under consideration in [54] (VAE, DCGAN, WGAN-GP, ADS-GAN) achieve more than $0.82$ in $A$ on simpler data (MNIST). This interpretation is strengthened by the results of a $C_T \gg 0$ which indicates that the model might even be underfitting and is not in a data copying regime. Qualitative results on the nearest neighbour search between training and synthetic data in Figure 1 further corroborate these quantitative results.

## 6 Conclusions

DiffInfinite offers a novel sampling method to generate large images in digital pathology. Due to the high-level mask generation followed by the low-level image generation, synthetic images contain long-range correlations while maintaining high-quality details. Since the model trains and samples on small patches, it can be efficiently parallelized. We demonstrated that the classifier-free guidance can be extended to a semi-supervised learning method, expanding the labelled data feature space with unlabelled data. The biological plausibility of the synthetic images was assessed in a survey by 10 domain experts. Despite their training, most participants found it challenging to differentiate between real and synthetic data, reporting an average low confidence in their decisions. We found that samples from DiffInfinite can help in certain downstream machine learning tasks, on both in- as well as out-of-distribution datasets. Finally, authenticity metrics validate DiffInfinite's capacity to generate novel data points with little similarity to the training data which is beneficial for the privacy preserving use of generative models in medicine.

---

[4]We use https://github.com/marcojira/fls from [81] to calculate both scores.

# 7 Acknowledgements

We would like to acknowledge our team of pathologists who provided valuable feedback in and outside of the conducted survey - special thank you to Frank Dubois, Niklas Prenissl, Cleopatra Schreiber, Vitaly Garg, Alexander Arnold, Sonia Villegas, Rosemarie Krupar and Simon Schallenberg. Furthermore, we would like to thank Marvin Sextro for his support in the analyses. This work was supported by the Federal Ministry of Education and Research (BMBF) as grants [SyReal (01IS21069B)]. RM-S is grateful for EPSRC support through grants EP/T00097X/1, EP/R018634/1 and EP/T021020/1, and DI for EP/R513222/1. MA is funded by Dotphoton, QuantIC and a UofG Ph.D. scholarship.

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
