# A Histological Dataset

Table 3: Details of the histological dataset

| | Histological dataset |
|---|---|
| **Number of whole slide images** | 41 |
| **Image type** | H&E-stained whole slide images |
| **Whole slide image size** | $\sim 100,000 \times 100,000$ |
| **Magnification** | 40x |
| **Image scanner** | Aperio scanner |
| **Number of annotation categories** | 40 |
| **Annotation distribution** | 37% Carcinoma, 36% Stroma, 3.5% Necrosis, 23.5% Other |
| **Resolution** | 0.5 microns per pixel |
| **Number of patches (image training)** | 4,781 labelled + 255,643 unlabelled |
| **Patch size** | $512 \times 512$ px |
| **Train/Test split** | 90/10 stratified by annotation categories |
| **Number of patches (large mask training)** | $1,183$ |
| **Patch size** | $2048 \times 2048$ px |

The real-world data used for training the generative model consisted of 41 high-resolution Hematoxylin and Eosin (H&E)-stained whole slide images of lung tissue biopsies from different cancer patients. These images were evenly split between cases diagnosed with adenocarcinoma of the lung and squamous cell carcinoma, representing the two most common sub-types in lung cancer. The images were scanned on an Aperio scanner at a resolution of 0.25 microns per pixel (40x). Different classes used for conditioning were annotated digitally by a pathologist using an apple pencil with the instruction to clearly demarcate boundaries between tissue regions. The pathologist could choose from a list of 40 distinct annotation categories, aiming to cover all possible annotation requirements. 37% of the annotations belonged to the Carcinoma category, 36% to Stroma, 3.5% to Necrosis and the remaining 23.5% to other smaller categories summarized as Other. All data handling was performed in strict accordance with privacy regulations and ethical standards, ensuring the protection of patient information at all times. For training the diffusion model, we utilized a patch dataset derived from expert annotations. In total, the dataset contained 4,781 patches of size $512 \times 512$ px. The dataset was split into train/ test sets with a ratio of 90/ 10, stratified by annotation categories. This test split was used for the generative model as well as to evaluate the downstream task. We also tiled the slides with size $2048 \times 2048$ from the same annotations, extracting 1,183 patches. These masks are used for training the mask generative model.

## A.1 Downstream task datasets

For the downstream tasks we utilized additional internal and external datasets to assess the predictive performance of our models.

Table 4: Details of downstream task datasets

| Dataset | Indication | Patch size | mpp | Number of patches |
|---|---|---|---|---|
| IH1 | lung | 512 | 0.5 | 3.7 K |
| IH2 | lung | 512 | 0.5 | 0.7 K |
| IH3 | lung | 512 | 0.5 | 2.8 K |
| NCT | colorectal | 224 | 0.5 | 100.0 K |
| CRC | colorectal | 224 | 0.5 | 7.0 K |
| PCam | lymph nodes | 96 | 0.972 | 327.0 K |

# B  Survey

The survey was sent out to 10 pathologists with varying years of experience. Fig.4 shows the setup of the survey. The presented images were shown in randomized order.

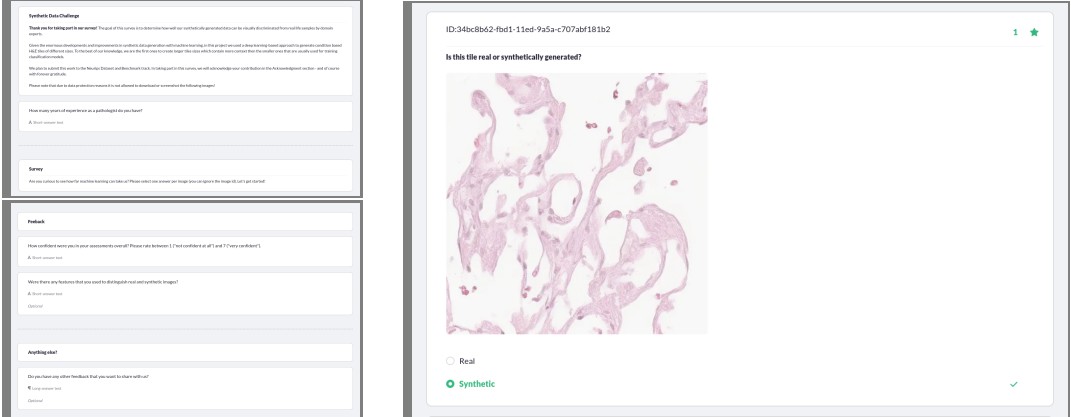

Figure 4: Survey interface for the domain expert assessment of real versus synthetic data.

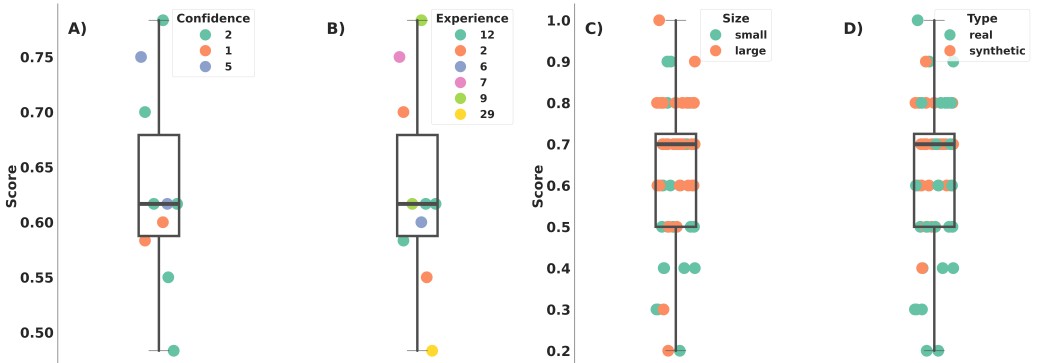

Figure 5: Results of the survey. Left: Accuracy per pathologist, color-coded by subjective confidence level (**A**) and years of experience (**B**). Right: Average accuracy across pathologists for each image patch, color-coded by path-size (**C**) and veracity (**D**).

## C Metrics for Data Assessment

In this section we provide the definitions of the metrics used in Section 5 to assess the fidelity and degree of memorization of DiffInfinite. Following the notation of Section 4.1, denote by $X_r \sim \mathcal{X}_r$ the real data distribution and by $X_g = \Psi(\hat{Y}) \sim \mathcal{X}_g$ the distribution from which the generative model samples. For the quantitative evaluation of the quality and the coverage of the data generated by DiffInfinite we use

**Improved recall and improved precision [67]**  A pre-trained classifier[5] maps the samples into a high-dimensional feature space resulting in the feature vectors $\Phi_r$ and $\Phi_g$. For $\Phi \in \{\Phi_r, \Phi_g\}$ denote by $NN_k(\phi', \Phi)$ the $k$th nearest feature vector of $\phi'$ from set $\Phi$ and define the binary function

$$f(\phi, \Phi) = \begin{cases} 1, & \text{if } \|\phi - \phi'\|_2 \leqslant \|\phi' - NN_k(\phi', \Phi)\|_2 \text{ for at least one } \phi' \in \Phi \\ 0, & \text{otherwise} \end{cases} \tag{10}$$

that identifies whether a given sample $\phi$ is within the estimated manifold volume of $\Phi$ corresponding to $NN_k$.    To measure the similarity of $\Phi_g$ to the estimated manifold of the real images, define improved precision (IP) by

$$\text{precision}(\Phi_r, \Phi_g) = \frac{1}{|\Phi_g|} \sum_{\phi_g \in \Phi_g} f(\phi_g, \Phi_r) \tag{11}$$

and to measure the similarity of $\Phi_r$ to the estimated manifold of the generated images, define improved recall (IR) by

$$\text{recall}(\Phi_r, \Phi_g) = \frac{1}{|\Phi_r|} \sum_{\phi_r \in \Phi_r} f(\phi_r, \Phi_g). \tag{12}$$

The rate of DiffInfinite to innovate a new sample is approximated by the

**Authenticity score [54]**  For the definition of the authenticity score $A \in [0, 1]$, assume that the probability measure $\mathbb{P}_g$ corresponding to $\mathcal{X}_g$ is a mixture of the probability measures

$$\mathbb{P}_g = A \cdot \mathbb{P}'_g + (1 - A) \cdot \delta_{g,\epsilon}, \tag{13}$$

where $\mathbb{P}'_g$ characterizes the generative distribution, excluding synthetic samples that are duplicates of training samples and $\delta_{g,\epsilon} = \delta_g * \mathcal{N}(0, \epsilon^2)$ is the noisy distribution over training data implied by an unknown discrete probability measure $\delta_g$ placing probability mass on each data point used for training.

To test DiffInfinite for data-copying we compute the

**$C_T$ score [80]**  For a set of training images $\mathcal{D}_{train} = \{x_1, ..., x_k | x_i \sim \mathcal{X}_r\}$ and $y \in \mathbb{R}^{KD}$ define the distance measure $d(y) = \min_{x \in \mathcal{D}_{train}} \|x - y\|_2^2$. Denote by $L(\mathcal{V})$ the one dimensional distribution $d(V)$ of any random variable $V \sim \mathcal{V}$ with the same instance space as $\mathcal{X}_r$. For the test set of the real data $\mathcal{D}_{test} = \{y_1, ..., y_n \mid y_i \sim \mathcal{X}_r\}$, define the fraction $P_n(\pi) = |\{y \in \mathcal{D}_{test} \mid y \in \pi \in \Pi\}| / n$ of test points in cell $\pi \in \Pi$, where $\Pi$ is a partition of $\mathbb{R}^{KD}$ resulting from applying the $k$-means algorithm on $\mathcal{D}_{train}$. Similar for a set of generated images $\mathcal{D}_{gen} = \{\hat{x}_1, ..., \hat{x}_m\}$ sampled from $\mathcal{X}_g$, define the fraction $Q_n(\pi)$ of generated samples in cell $\pi \in \Pi$. Denote by $Z_U$ the $z$-scored Mann-Whitney $U$ statistic from Section 3.1 of [80] with $L_\pi(\mathcal{D}) = \{d(x) \mid x \in \mathcal{D}, \pi \in \Pi\}$ for $\mathcal{D} \in \{\mathcal{D}_{test}, \mathcal{D}_{gen}\}$ and let $\Pi_\tau$ be the set of all cells in $\Pi$ for which $Q_m(\pi) \geqslant \tau$ holds true. The $C_T$ score is finally defined as the average

$$C_T(P_n, Q_m) = \frac{\sum_{\pi \in \Pi_\tau} P_n(\pi) Z_U\big(L_\pi(P_n), L_\pi(Q_m); T\big)}{\sum_{\pi \in \Pi_\tau} P_n(\pi)}. \tag{14}$$

across all cells represented by $\mathcal{X}_g$.

---

[5]We use the pre-trained VGG-16 classifier from https://github.com/blandocs/improved-precision-and-recall-metric-pytorch.

Table 5: Quantitative memorization metrics for the variants of DiffInfinite described in Section 5.1. For consistency, we consider all methods from Table 1 in our evalution, including the comparison to DiffCollage. For the methods that output a large image of size 2048 we consider the *tiled* patches resulting in 16 patches per large image and the *resized* image resulting in 200 images of size $512 \times 512$.

|  | $A\uparrow$ | | $C_T \downarrow\uparrow$ | |
|---|---|---|---|---|
|  | *tiled* | *resized* | *tiled* | *resized* |
| DiffCollage | **0.89** | 0.97 | 11.02 | **7.00** |
| DiffInfinite (a) | 0.86 | - | 4.99 | - |
| DiffInfinite (b) | 0.86 | 0.97 | **3.29** | 8.11 |
| DiffInfinite (c) | 0.86 | **0.98** | 9.61 | 11.56 |
| DiffInfinite (b) & (c) | 0.87 | 0.95 | 5.31 | 10.96 |

# D  Data Samples

**Mask-image pairs**  In Fig. 6, we show the control on the mask-image generation for $512 \times 512$ patches. The *Unknown* class corresponds to pixels which were not annotated due to a sparse annotation strategy. The images show that the cross-attention layer controls mask conditioning effectively. As a proof of concept, we generated images at different scales ($512 \times 512, 1024 \times 1024, 2048 \times 2048$) with a simple squares mask (see Fig. 7). In Figure 8, we see that for the small masks of size $512 \times 512$, the frequency of labels in the real masks are reproduced well by the generated masks. For the large masks of size $2048 \times 2048$, the labels that occur most frequently in the real masks are underrepresented in the generated masks, while all other labels are overrepresented in the generated masks.

**Random patch advantages**  Sampling with the random patch (RP) method leads to several benefits compared to the sliding windows (SW) approach (see Fig. 11). First, the sliding window method starts from the centre of the image and outpaints in four directions. As a consequence, the model needs to condition on previously generated areas, leading to blurriness on the farther pixels. With the random patch method, every area is conditioned only on its neighbour, avoiding error propagation. Moreover, while SWs have only information on the closest neighbour, RPs consider long-range correlations. On every diffusion step, we have every possible overlap between near patches, extending correlation lengths to twice the diffusion model output size. Furthermore, this random overlap avoids any tiling effect.

**Inpainting**  Using the segmentation images and masks of the test set, we inpainted the annotated areas with the same corresponding class (see Fig. 9). We show that the model generates new content respect to the real one. We run the same experiment by inpainting one area with different classes (see Fig. 10). Keeping the same seed, we show how the generation changes while $\omega$ increases. By increasing $\omega$, we enhance the diversity at the cost of losing some conditioning.

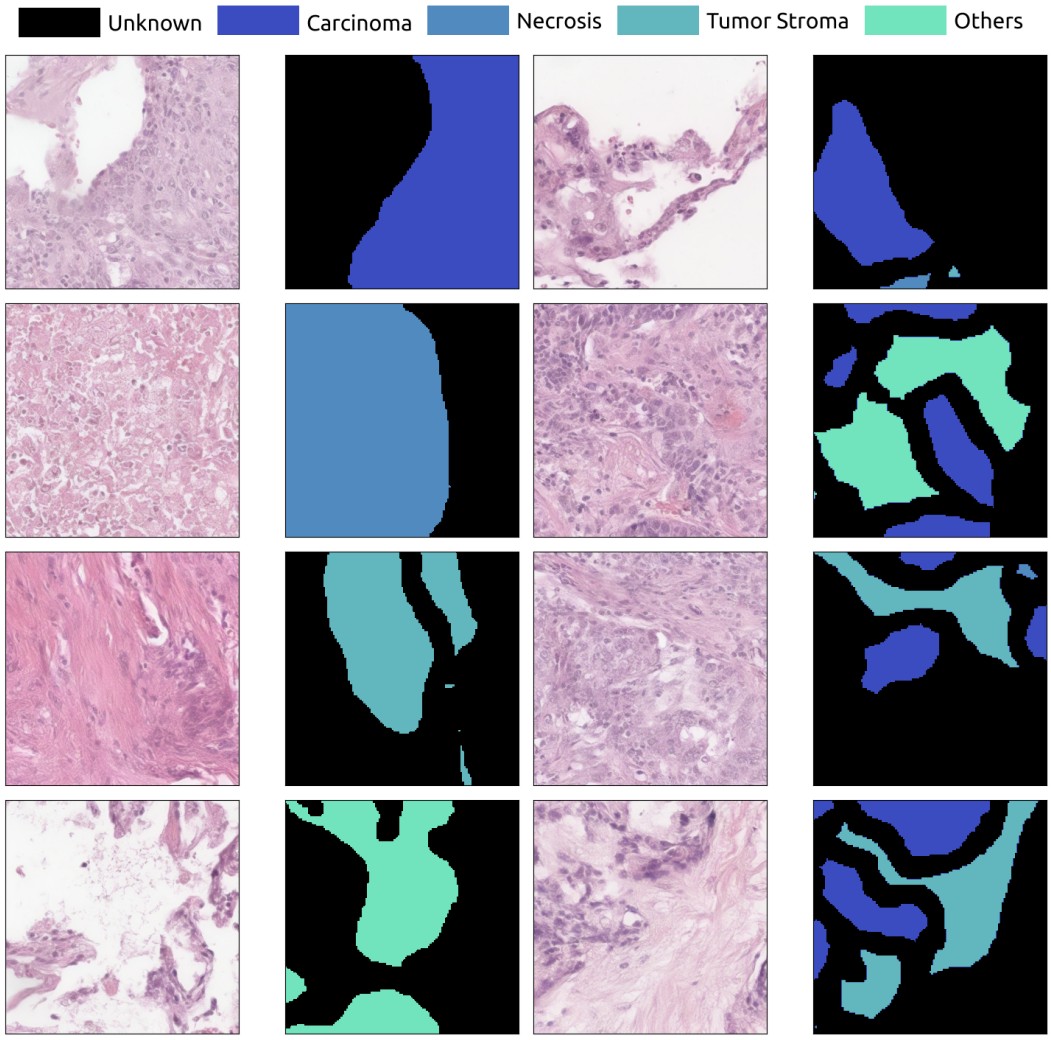

Figure 6: Generated images conditioned on the synthetic segmentation masks.

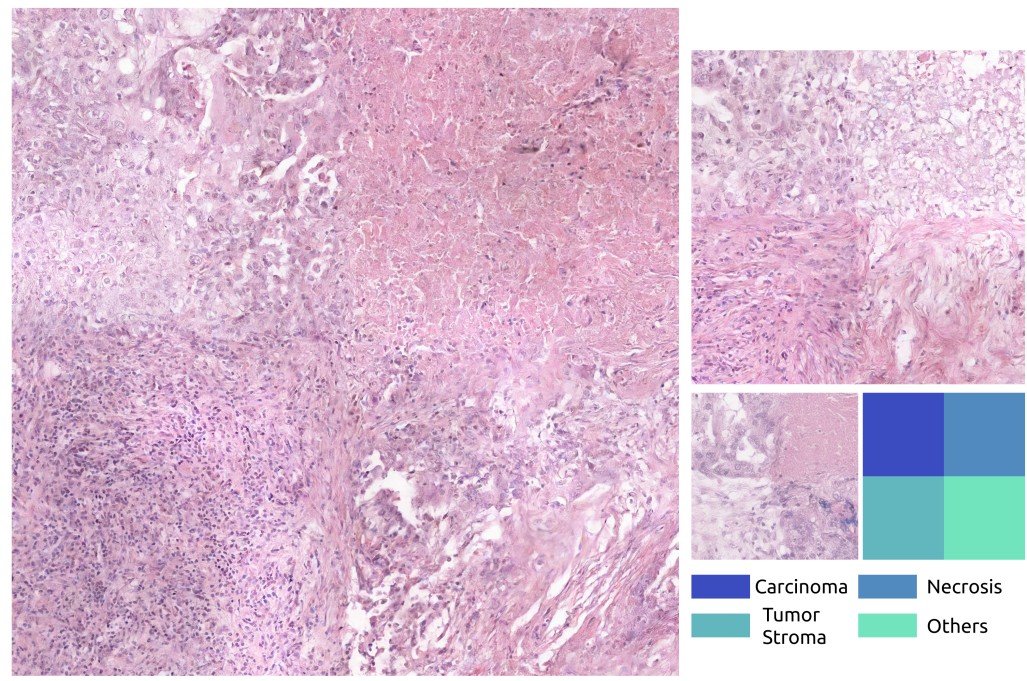

Figure 7: Conditioning visualization. All the images are conditioned with the squared mask shown. Left) $2048 \times 2048$ image. Top-Right) $1024 \times 1024$ image. Bottom-Right) $512 \times 512$ images.

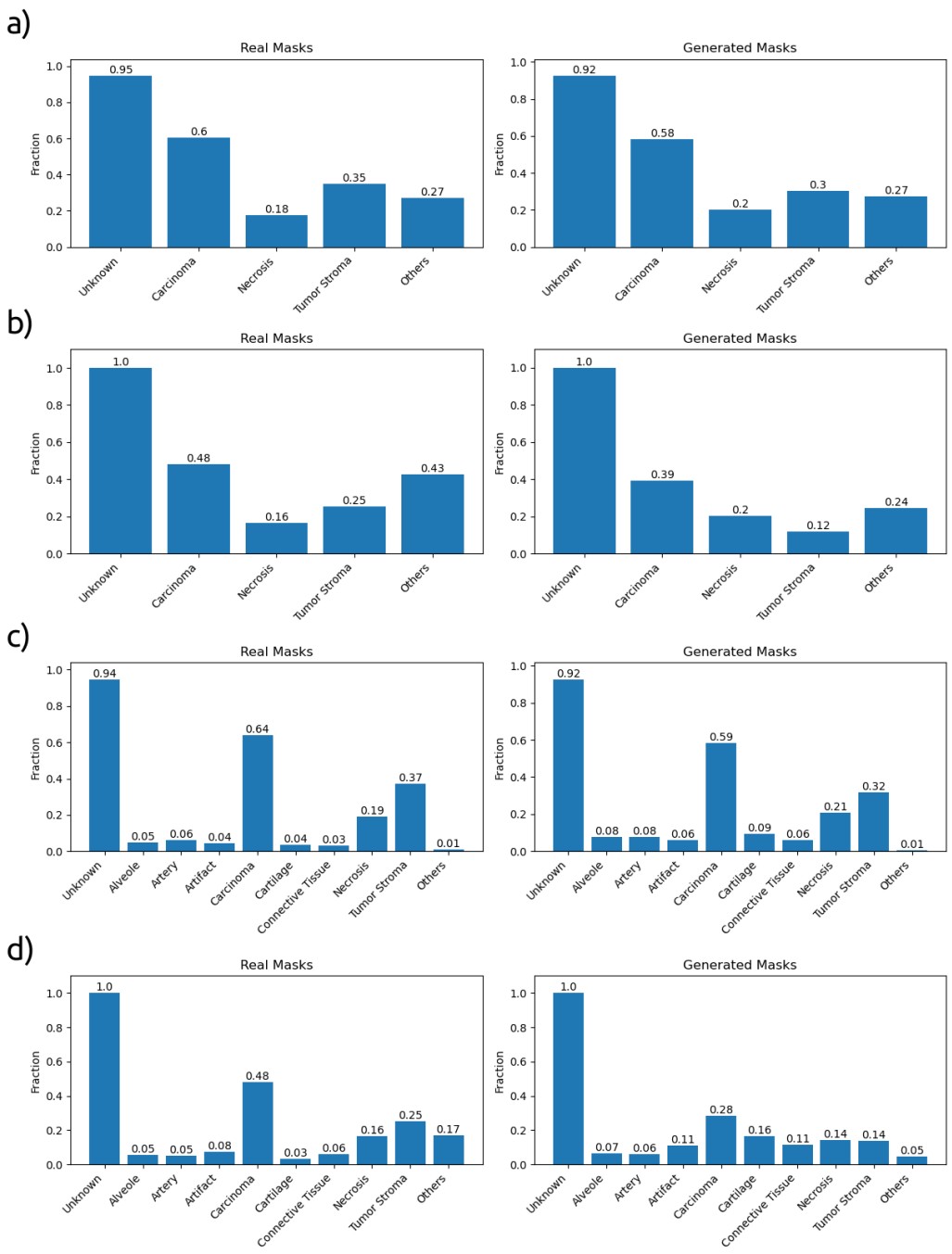

Figure 8: Fraction of label appearance in the segmentation masks with 5 classes in a,b) and 10 in c,d). Fractions estimated over a) 4205 real masks of size $512 \times 512$ and 20719 generated masks, b) 1183 real masks of size $2048 \times 2048$ and 22705 generated masks, c) 4205 real masks of size $512 \times 512$ and 22604 generated masks, d) 1183 real masks of size $2048 \times 2048$ and 22560 generated masks.

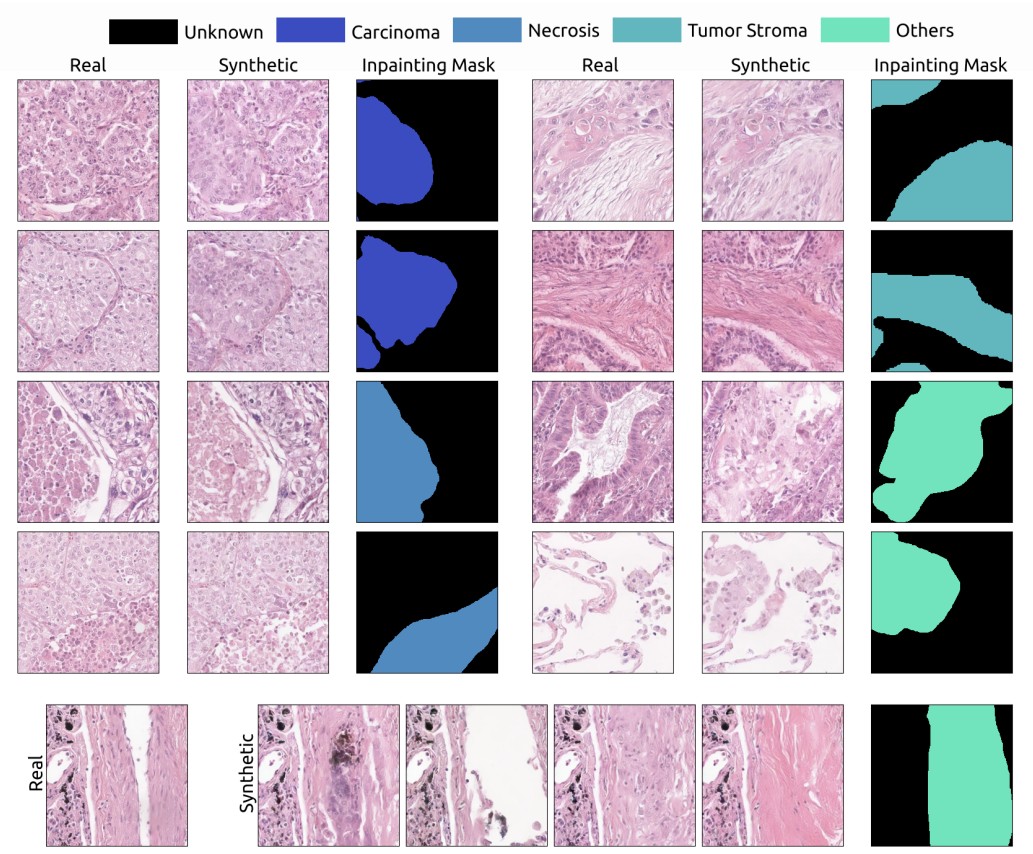

Figure 9: Inpainting test data with the corresponding label. Top) Inpainting for different labels. Bottom) Different inpainted synthetic areas for the same mask.

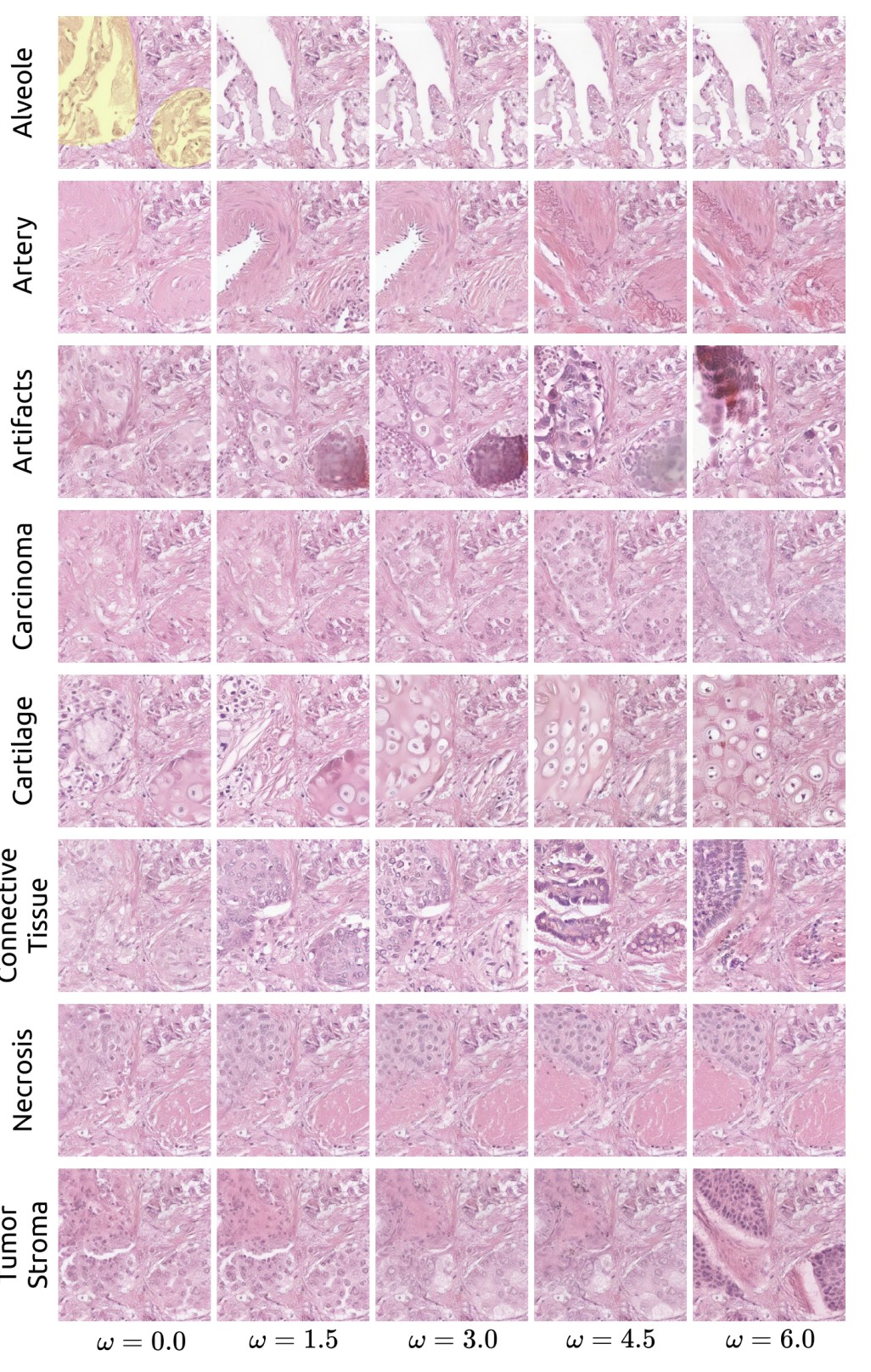

Figure 10: Proof of concept with inpainting. We inpainted the same base image with different classes and different strengths of conditioning (small $\omega$ corresponding to less diversity). The corresponding inpainting mask is displayed as an overlay on the top left patch (in yellow).

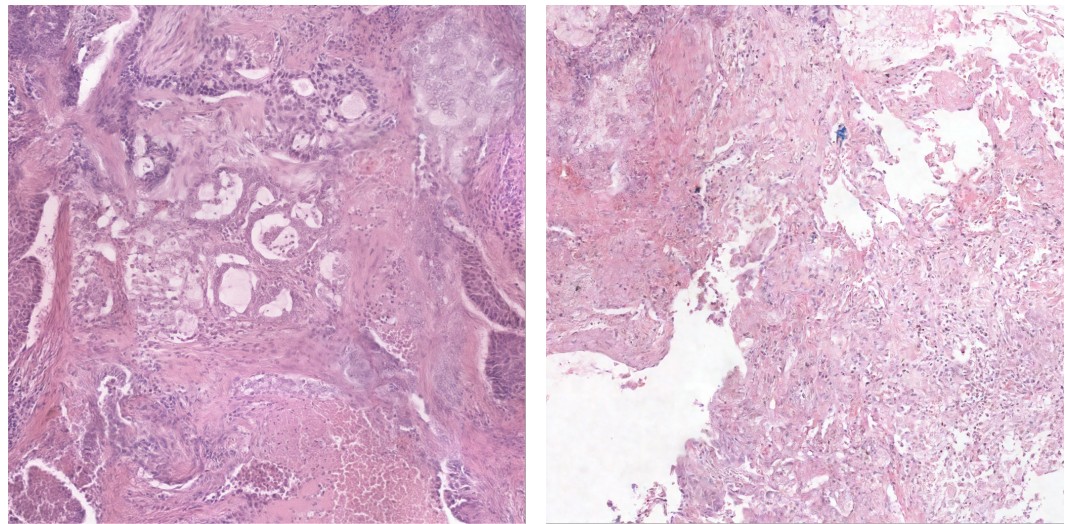

Figure 11: Comparison of different methods to generate large images (2048 × 2048). Left) DiffCollage image generation using the grid graph [48]. Right) DiffInfinite (ours) image generation using the proposed random patch sampling.

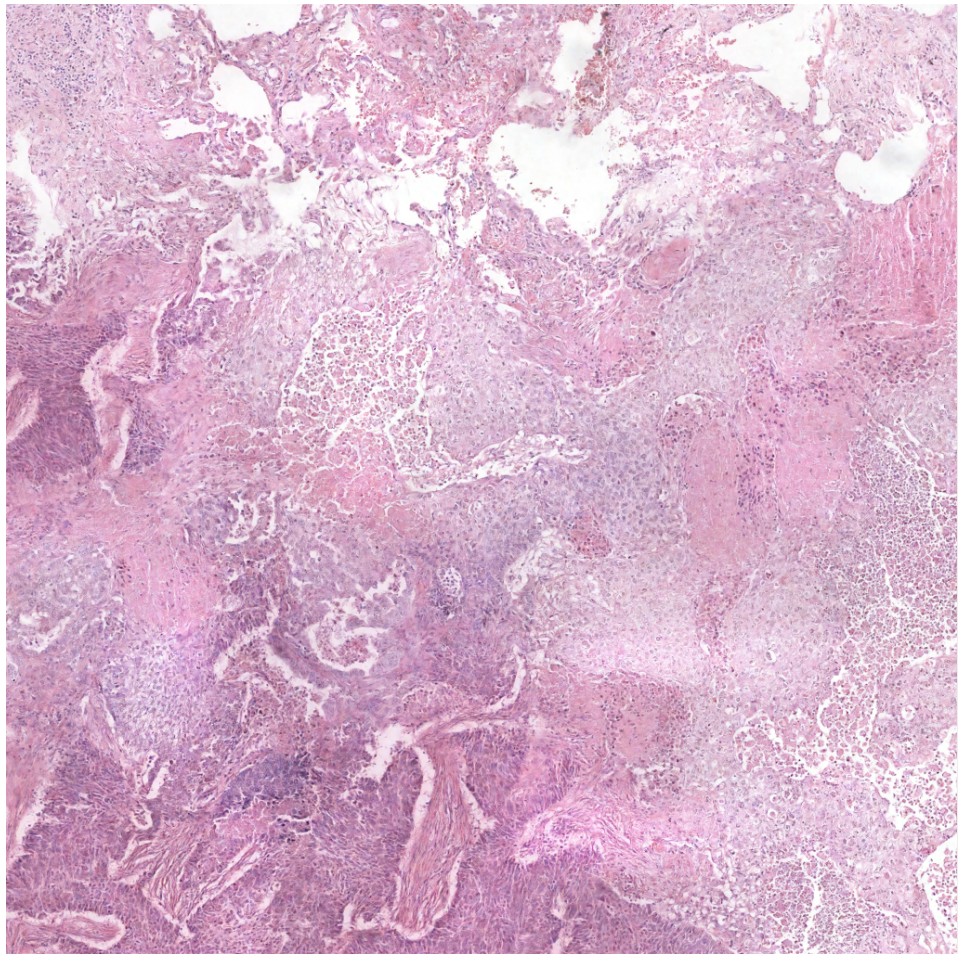

Figure 12: Large-content synthetic image with a size of 4096x4069 pixels.

# E   Training Details

**Training on images**  The core model used in the diffusion process is a U-Net[6]. Every U-net's block is composed of two ResNet blocks, a cross-attention layer and a normalization layer. On each ResNet block, we feed the output $x^l$ of the previous block $l$, the time $t$ and the label $c_i$. The cross-attention mask is performed using the mask corresponding to the label $c_i$ as query and the input $x^l$ as key and value.

**Training on masks**  We replace the cross-attention with a linear self-attention layer for mask generation. Here, the model is conditioned with binary labels $\{0, 1\}$, where 0 corresponds to adenocarcinoma and 1 corresponds to squamous cell carcinoma. The masks of size $512 \times 512$ is first downsampled to size $1 \times 128 \times 128$. We stack the downsampled mask to the size $(3, 128, 128)$ to make it compatible with a pre-trained VAE[7]. We repeated the same training for the larger masks $2048 \times 2048$, downsampling them to $128 \times 128$ as well.

Table 6: Details of the parameters used for training

| Model parameters image generation | | Model parameters mask generation | |
| --- | --- | --- | --- |
| **Image $X$ shape** | (3,512,512) | **Mask $M$ shape** | (3,128,128) |
| **Latent $Y$ shape** | (4,64,64) | **Latent $Y$ shape** | (4,16,16) |
| **VAE** | stabilityai/stable-diffusion-2-base | **VAE (repo id)** | stabilityai/stable-diffusion-2-base |
| **Num classes** | 5 and 10 | **Num classes** | 2 |
| **Loss** | L2 | **Loss** | L2 |
| **Diffusion steps** | 1000 | **Diffusion steps** | 1000 |
| **Training steps** | 250000 | **Training steps** | 100000 |
| **Sampling steps** | 250 | **Sampling steps** | 250 |
| **Heads** | 4 | **Heads** | 4 |
| **Heads channels** | 32 | **Heads channels** | 32 |
| **Attention resolution** | 32,16,8 | **Attention resolution** | 32,16,8 |
| **Num Resblocks** | 2 | **Num Resblocks** | 2 |
| **Probability $p_{unc}$** | 0.5 | **Probability $p_{unc}$** | 0.5 |
| **Batch size** | 128 | **Batch size** | 64 |
| **Number of workers** | 32 | **Number of workers** | 1 |
| **GPUs Training** | 4 NVIDIA GeForce RTX 3090, 24Gb each | **GPUs Training** | 2 Ampere A100, 40Gb each |
| **GPUs Inference** | 1 NVIDIA GeForce RTX 3090 | **GPUs Inference** | 1 NVIDIA GeForce RTX 3090 |
| **Training time** | $\sim$ 1 week | **Training time** | $\sim$ 4 hours |
| **Optimizer** | Adam | **Optimizer** | Adam |
| **Scheduler** | OneCycleLR(max lr=1e-4) | **Scheduler** | OneCycleLR(max lr=1e-4) |

---

[6]Baseline, https://github.com/lucidrains/classifier-free-guidance-pytorch
[7]https://huggingface.co/stabilityai/stable-diffusion-2

# F   Sampling Details

**Mask cleaning**   The diffusion model samples a latent mask in the VAE's latent space. After mapping the latent mask back to the pixel space we average over the channels to have a mask with one channel and round the pixel values to the integers $\{0, 1, ..., num\_values\}$. Since we note some boundary artifacts between regions of different values we first apply a method from skimage [8] to find these boundary artifacts and replace it by $0$, corresponding to unknown area. Before resizing the mask to the full size, we apply a minpooling operation to erase labelled regions of small magnitude and replace it as well with unknowns.

**Hann windows decoding**   After the diffusion model samples $Z$ in the VAE's latent space, the latent variable $Z$ needs to be decoded into the pixel space. However, due to computational constraints, it is not feasible to decode $Z$ all at once. Therefore, we tile it into smaller patches. Decoding smaller patches would introduce tiling effects. In order to reduce edge artifacts, we used an overlapping window method using Hann windows as weights [82]. In Fig. 13, we tile the image in four different configurations such that the edges and corners are overlapping, and then we perform a weighted sum over the upsampled outputs.

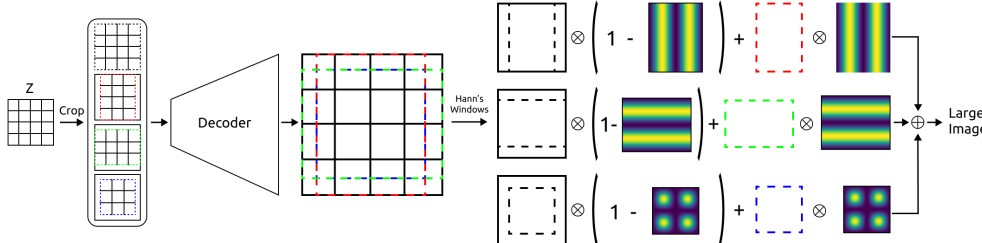

Figure 13: Hann window overlapping illustration.

---

[8]https://scikit-image.org/docs/stable/api/skimage.segmentation.html