# OpenReview forum: "DiffInfinite: Large Mask-Image Synthesis via Parallel Random Patch Diffusion in Histopathology"
_NeurIPS.cc/2023/Track/Datasets_and_Benchmarks — NeurIPS 2023 Datasets and Benchmarks Spotlight_

### Official Review · Reviewer_hrbm · 2023-07-10
**A review**

**Rating:** 8
**Confidence:** 3
**Clarity:** Paper is well written.

**Strengths:**

* Whole Slide Images (WSIs) are overall quite costly to gather, and the community could greatly benefit from synthetic WSIs if they are created appropriately.
* The sampling method they proposed seems to both avoid the artifacts resulting from tiling observed in earlier works (Zhang et. al. https://arxiv.org/abs/2303.17076), and it doesn't have the constraints that sliding sampling outpainting methods have, which can only be paralleled 2-4 times.
* While it's highly risky to use patient data when training a generative model, it appears that the authors conducted experiments to address potential memorization issues. The generated samples seem to be diverse enough compared to the ones seen in the training data.

**Additional Feedback:**

Typo at random patch diffusion (row 184):

should be from $c_1$ to $c_N$ I think.

**Correctness:**

Interestingly, the submission is neither a dataset nor a benchmark, making it difficult to categorize. However, they evaluated the generated synthetic samples through both human evaluation and a downstream segmentation task, which seems valid. In the "Opportunities for Improvement" section, I noted how beneficial it would be if they could evaluate the model on a downstream classification task instead of a segmentation task.

**Documentation:**

* In the appendix, the authors shared some information about their training data and model hyperparameters. However, they did not mention the type of hardware used, the duration of training, the batch size - in short, how affordable the model would be if someone wanted to train it from scratch, which is a limitation.
* Additionally, the authors did not provide any URLs for reviewer access to view the generated synthetic samples, nor did they make the model available for reviewer access. Although they promised to share the code upon acceptance, it would have been beneficial to have access to this information earlier during the review process.

**Ethics:**

I do not have any ethical concerns.

**Limitations:**

The authors adequately addressed the limitations and potential negative societal impact of their work.

**Opportunities For Improvement:**

* Nothing related to the model is provided, and no dataset URL is given for reviewing. In the checklist, they promise to share the code for generating synthetic images. Considering their training data, it's hard to believe that a model trained on just 41 WSIs (which at the patch level is still substantially smaller than what standard diffusion models are trained on in natural image settings) can produce synthetic images that are indistinguishable and are showing substantial differences to the training data (not memorized). Having access to the model during the review process for experimentation would have been beneficial.

* “When pathologists were asked which features they used to differentiate real from synthetic images, they reported features like inconsistent patterns, overly homogeneous chromatin in synthetic nuclei, peculiarities in cellular and intercellular structures, and aesthetic elements as discriminating factors. These seemed to be especially pronounced in tumorous regions where sometimes the tissue architecture appeared exaggerated, the transition to stroma or surrounding tissue was too abrupt and some cells lacked distinguishable nucleoli or cytoplasm.” In my opinion, this indicates that the model doesn't do a good job generating regions of interest where generation "accuracy" matters most. I presume it's easy to learn "background" patches, but learning rarer tumorous patches might be more challenging. Again, having access to their model, or some synthetic samples during reviewing, would have been helpful to test this.

* Additionally, the quantitative evaluation is based on segmentation. I think patch-level classification would be more interesting because the segmentation masks are already provided to the model during training, so it would be intriguing to see how much signal the generated patches carry in classification tasks. Perhaps the authors could create a whole slide image, then patchify it, train a model, and perform zero-shot evaluation on basic datasets like PatchCamelyon (https://patchcamelyon.grand-challenge.org), NCK-CRC (https://zenodo.org/record/1214456), etc. (I am very willing to increase my rating to 7 If this will be conducted.)

* The inter-rater variance in the human evaluation part is somewhat concerning. Instead of considering years of experience and confidence, it might be useful to know how much time pathologists spent before deciding whether the displayed image is synthetic or not.

* The authors did not provide any specifics regarding the hardware they used, making it challenging to estimate the potential cost of training a similar model from scratch.

**Relation To Prior Work:**

The authors have discussed several related works and it is clearly discussed how their work differs.

**Summary And Contributions:**

Overall, I appreciate the idea! Initially, they train a diffusion model, $E_\theta$,  conditioned by class labels found within the given sub-patch. The diffusion model learns to generate these subpatches individually, then combines all of them to generate a patch. The same diffusion model, $E_\theta$, is then used to generate Whole Slide Images (WSI), by using a hierarchical mapping of the desired WSI (A sketch of the cellular arrangements) which guides the model managing long-range correlations while outpainting. They evaluated the generated synthetic WSIs on a downstream segmentation task and validated the samples. Also 10 human doctors evaluated the samples by deciding whether or not the samples are synthetic or not and reported humans cannot easily distinguish the real patches from the synthetic ones again showing the validity.

*The semi-supervised, classifier-free guidance can be seen as a minor innovation.

* The sampling method they proposed seems both avoiding artifacts due to tiling that was seen at earlier works (Zhang et. Al. ) and they don’t have constraints that sliding sampling outpainting methods has which can only process 2-4 times in parallel

---

> ### Author Response · Authors · 2023-08-22
>
> Dear Reviewer hrbm,
>
> thank  you  for  your  detailed  and  constructive feedback on our manuscript.  Please find our responses to your  comments below:
>
> 1. > Model and Dataset Access
>    >
>
> We appreciate your feedback regarding the necessity of having access to the model and dataset details.  We have taken this feedback into account and have now added the link to our repository, containing all necessary resources, to our paper.  Please find all the relevant links below:
>
> - Algorithm code: [https://github.com/diffinfinite/diffinfinite](https://github.com/diffinfinite/diffinfinite)
> - Model weights: [https://github.com/diffinfinite/diffinfinite/blob/master/download.py](https://github.com/diffinfinite/diffinfinite/blob/master/download.py) (can be downloaded via script)
> - Set of example images: [https://github.com/diffinfinite/diffinfinite/tree/master/results](https://github.com/diffinfinite/diffinfinite/tree/master/results)
> - Interactive notebook to interact with the model: [https://colab.research.google.com/github/diffinfinite/diffinfinite/blob/master/main.ipynb](https://colab.research.google.com/github/diffinfinite/diffinfinite/blob/master/main.ipynb)
>
> 2. > Size of training data
>    >
>
> Thank you for pointing out that a training set of 41 whole slide images seems very small in comparison to other diffusion model trainings. We would like to clarify that given the gigapixel size of whole slide images and the semi-supervised nature of our generative model, the training set of our generative model consisted of 5,000 annotated patches and 250,000 unlabeled patches. We’ve updated Table 4 in Appendix A to elucidate the specific count of patches, both labeled and unlabeled. Furthermore, both qualitative (pathologist survey) and quantitative (authenticity, quality, and diversity metrics, downstream task) assessments demonstrate the quality and diversity of the generated images.  While this work showcases that the method is valuable even in relatively small data regime, we agree that with the modular and scalable nature of the model architecture there is large potential in including more and diverse WSIs in the future.
>
> 3. > Differentiation of Synthetic vs Real by Pathologists
>    >
>
> We respect your assessment regarding the qualitative differences observed by pathologists in our synthetic images.  It’s crucial to note that our general feedback from pathologists was overwhelmingly positive. The nuances highlighted were mentioned when specifically asked to offer an explanation for possible features to distinguish real and synthetic images. The mentioned features, which were used as differentiating factors, came into play only when the pathologists were specifically asked to critique. As the results suggest, generally pathologists were not able to successfully differentiate between the images. We’ve now updated Section 5.2 to accentuate that these remarks underscore minor deviations rather than prevalent discrepancies.

---

> > ### Author Response · Authors · 2023-08-22
> >
> > 4. > Quantitative Evaluation Basis and Patch-level Classification
> >    >
> >
> > We acknowledge the significance of integrating a patch-level classification for a comprehensive evaluation and have followed your suggestion.
> > The main reason for not considering it before was that we wanted to assess the inherent ability of our model to generate segmentation masks to condition the image generation as a strength of our model. Being able to generate labeled segmentation data to train a segmentation model is very valuable since good segmentation annotations are time consuming and costly to acquire. To further substantiate our claims, we’ve enriched our results section with additional segmentation outputs for reference. Concurring with your feedback on the value of a classification task, we’ve now incorporated supplementary classification experiments, documented in Section 5.3. We added both results on proprietary datasets that were also used to evaluate the segmentation model as well as on the public datasets you mentioned trying to test our model as broadly as possible.
> >
> > |                   | **IH1**  | **IH2**  | **IH3**  | **NCT-100K** | **CRC-7K** | **PCam-327K** |
> > | ----------------- | -------------- | -------------- | -------------- | ------------------ | ---------------- | ------------------- |
> > | Trained Real      | 0.85           | 0.73           | 0.60           | 0.87               | **0.86**   | 0.60                |
> > | Trained Synthetic | 0.75           | **0.75** | **0.70** | 0.77               | 0.72             | **0.68**      |
> > | Trained Augmented | **0.85** | 0.73           | 0.64           | **0.87**     | 0.80             | 0.63                |
> >
> > Zero-shot evaluation of classifiers on internal (IH2 and IH3) and external data (NCT, CRC,PCam) with increasing components of data drift. Three classifiers were trained to perform binaryclassification for presence of adenocarcinoma on patch level. “Trained Real” was trained with realin-house (IH1) histo data that was used to train DiffInfinite. “Trained Synthetic” was trained withsamples from DiffInfinite, and “Trained Augmented” used both real and synthetic data. The publicexternal datasets comprise: NCT (224x224 pixels, 100K datapoints, 0.5 MPP, colorectal), CRC(224x224, 7K datapoints, 0.5 MPP, colorectal), PCam (PatchCamelyon) (96x96, 327K datapoints,0.972 MPP, lymph nodes).
> >
> > |                   | **IH2**  |
> > | ----------------- | -------------- |
> > | Trained Real      | 0.61           |
> > | Trained Synthetic | 0.47           |
> > | Trained Augmented | **0.71** |
> >
> > *New segmentation results: new baseline models for the segmentation task for better comparison of effect of synthetic data on segmentation model.*
> >
> > 5. > Inter-rater Variance in Human Evaluation
> >    >
> >
> > Thank you for pointing out the potential limitations associated with inter-rater variance. We would like to emphasize that the time allotted to pathologists mirrors what they typically have in a clinical setup. Hence, the evaluations are reflective of practical scenarios. We have edited section 5.2 to highlight that our analyses revealed no substantial correlation between the time expended by the pathologists and the accuracy of their evaluations.
> >
> > 6. > Hardware specifications
> >    >
> >
> > Pointing out the benefit of adding the hardware specifications of our experiments is a great suggestion. We have added additional information to  Table 5 in Appendix E that mentions hardware used for training the generative model as well as an approximate time the model training took.
> >
> > |                   | Image generation                     | Mask generation          |
> > | ----------------- | ------------------------------------ | ------------------------ |
> > | Number of workers | 32                                   | 1                        |
> > | GPUs Training     | 4 NVIDIA GeForce RTX 3090, 24Gb each | 2 Ampere A100, 40Gb each |
> > | GPUs Inference    | 1 NVIDIA GeForce RTX 3090            | 1 NVIDIA GeForce RTX     |
> > | Training time     | 1 week                               | 4 hours                  |

---

> > > ### Author Response · Authors · 2023-08-22
> > >
> > > 7. > Dataset and Benchmark Track
> > >    >
> > >
> > > We’d like to highlight that the call for papers explicitly includes ”data generators” and we  believe it is relevant to the Datasets & Benchmarks track as we identify and make contributions  towards addressing data-centric issues specific to histopatological imaging, namely: reducing  the need for costly manual annotation of whole slide images; minimising the usage of real  patient data in downstream machine learning tasks; and providing the ability to efficiently  generate synthetic images at resolution which matches real WSI much more closely than before.  Indeed, several papers addressing wider data-centric issues have been accepted in this track in previous years, including  https://openreview.net/forum?id=v3yM5zVzP4C, which similarly addresses a data-centric problem and then proposes an improved way of curating data, and  https://openreview.net/forum?id=zNQBIBKJRkd  which discusses trends in dataset and benchmark curation within the machine learning community and which received the best paper award in 2021. While we provide some synthetic data samples to  illustrate the quality of the generated data, our contribution goes beyond providing synthetic dataset. Our method and codebase, which we make publicly available, allows the community to efficiently generate any number of unique datapoints, while providing flexibility in terms of target resolution and spatial distribution, while at the same time, overcoming the data issues outlined above.
> > >
> > > We are grateful for the comprehensive review and valuable suggestions. We believe our revisions fully address the raised concerns, and we hope this enhanced version of our paper will be received favorably.
> > >
> > > In good spirits,
> > >
> > > The authors

---

> > > > ### Comment · Reviewer_hrbm · 2023-08-22
> > > > **Thanks for your very detailed feedback to my comments!**
> > > >
> > > > Thank you for taking the time to address my concerns and making your model available! I have raised my score to 8.

---

### Official Review · Reviewer_Wamo · 2023-07-19
**DiffInfinite: Large Mask-Image Synthesis via Parallel Random Patch Diffusion in Histopathology**

**Rating:** 7
**Confidence:** 4
**Correctness:** All claims made in regards to the pro…
**Clarity:** [Yes] It is clear to read.

**Strengths:**

- The paper is well written and easy to follow. Also, the proposed framework is well motivated as synthetic image generation in medical data is challenging, especially in histopathology images that exhibit complex tissue micro-environments.

- While synthetic generation of images in digital pathology is not new; the proposed framework is able to generate arbitrarily sized images (up to 4096px), compared to related works that were limited to small resolutions and exhibit artefacts. At the same time, a small-scale dataset was used for model training.

- The sampling procedure introduced in this work is both novel and interesting. Also, efficient inpainting with different conditioning parameters is validated to generate diverse images.

- The authors extensively evaluated the generated images via a survey with experienced pathologists, including a downstream segmentation task and metrics for anti-duplications.

**Additional Feedback:**

This work introduces several technical innovations for large scale image generation in digital pathology, but fails to contextualize the actual dataset contribution and it’s significance. The core contribution of the work is the framework, rather than a single unified benchmark to evaluate a task. In addition, the evaluation on downstream tasks (section 5.3) lacks several key details that may better help to assess the actual difficulty of the task.

**Documentation:**

Details regarding the dataset/collection used for model training are provided. Links to be released.

**Ethics:**

None.

**Limitations:**

- Significant technical contributions rather than dataset/benchmark. The submitted work’s core focuses largely on technical contributions rather than the dataset itself. Also, It is unclear how the proposed framework can be eventually used to benchmark tasks i.e., what is the envisioned tasks?

- While the reviewer appreciates the extensive evaluations on quality generated images, another concern is how fair evaluations can be guaranteed if only the model is released, and not the generated images.

- Details regarding the downstream tasks are unclear [Section 5.3]. It is unclear whether an in-house or public dataset was used. In addition, while the trained model reaches 64.16 F1 score, it is equally unclear if this performance is significant as their is no mention of baselines or an upper-bound.

- Missing evaluation with DiffCollage (Table 1).

**Opportunities For Improvement:**

[See Limitations]

**Relation To Prior Work:**

[-] Related sections are sufficient

**Summary And Contributions:**

This work introduces a novel sampling method in diffusion-based models for large image generation in digital pathology, mitigating annotation costs and facilating large-scale data generation for different downstream tasks.  In particular, a scalable hierarchical diffusion model coined ‘DiffInfinite’ is proposed to generate images that maintain long-range structural correlations while avoiding artifacts. As opposed to existing approaches, the diffusion process first generates synthetic segmentation masks that are subsequently used as conditions in the final image generation procedure, resulting in more fine-grained details. Aside from high quality image generation, the approach can also scale to arbitrary image sizes despite only training on smaller image regions of interest. The authors validate the framework’s ability to generate plausible images in a survey with expert pathologists, a downstream segmentation task, and introduce metrics for model scoring. Extensive ablations of key hyper-parameters reveal the framework can consistently generate diverse and quality images.

---

> ### Author Response · Authors · 2023-08-22
>
> Dear Reviewer Wamo,
>
> Thank you for your thoughtful and constructive feedback on our manuscript. Please find our responses to your comments below
>
> 1. > Significant technical contributions rather than dataset/benchmark
>    >
>
> We agree with your observation that our work emphasizes significant technical contributions. We would like to highlight that the call for papers explicitly includes "data generators" and we believe it is relevant to the Datasets \& Benchmarks track as we identify and make contributions towards addressing data-centric issues specific to histopatological imaging, namely: reducing the need for costly manual annotation of whole slide images; minimising the usage of real patient data in downstream machine learning tasks; and providing the ability to efficiently generate synthetic images at resolution which matches real WSI much more closely than before. Indeed, several papers addressing wider data-centric issues have been accepted in this track in previous years, including https://openreview.net/forum?id=v3yM5zVzP4C, which similarly addresses a data-centric problem and then proposes an improved way of curating data, and https://openreview.net/forum?id=zNQBIBKJRkd which discusses trends in dataset and benchmark curation within the machine learning community and which received the best paper award in 2021. We also agree with the reviewer that our work makes technical contributions, however, which opens up possibilities for curating new datasets. While we provide some synthetic data samples to illustrate the quality of the generated data, our contribution goes beyond providing synthetic dataset. Our method and codebase, which we make publicly available, allows the community to efficiently generate any number of unique datapoints, while providing flexibility in terms of target resolution and spatial distribution, while at the same time, overcoming the data issues outlined above.
>
> 2. > Fair evaluations and data release
>    >
>
> We appreciate your concerns regarding the evaluations. The executable code is now provided with the updated manuscript. In addition, we provide an interactive notebook to test and run the model as well as a set of data samples. All artifacts can be anonymously accessed at
>
> - Algorithm code: [https://github.com/diffinfinite/diffinfinite](https://github.com/diffinfinite/diffinfinite)
> - Model weights: [https://github.com/diffinfinite/diffinfinite/blob/master/download.py](https://github.com/diffinfinite/diffinfinite/blob/master/download.py) (can be downloaded via script)
> - Set of example images: [https://github.com/diffinfinite/diffinfinite/tree/master/results](https://github.com/diffinfinite/diffinfinite/tree/master/results)
> - Interactive notebook to interact with the model: [https://colab.research.google.com/github/diffinfinite/diffinfinite/blob/master/main.ipynb](https://colab.research.google.com/github/diffinfinite/diffinfinite/blob/master/main.ipynb)
>
> 3. > Details regarding the downstream tasks are unclear [Section 5.3]
>    >
>
> Regarding the experiments, we updated section 5.3, incorporating additional experiments for classification and segmentation tasks across in-house and out-of-distribution datasets. We further describe the outcome of this validation. In particular, we trained a model using the same real data patch dataset which was also employed for training the diffusion model. To assess the model’s ability to generalize to a distinct cohort of lung patients from a different hospital, we utilized an internal dataset of 13 additional whole slide images. This allowed us to comprehensively evaluate and compare performance between models trained on both real data and on synthetic data, as outlined in the updated Section 5.3. While it remains true that the performance of the model exclusively trained on synthetic data continues to lag behind the model trained exclusively on real data in some domains, we present evidence that supplementing the real data with synthetic data enhances the model's capacity to generalize to the external dataset from a different distribution.
>
> |                   | **IH2**  |
> | ----------------- | -------------- |
> | Trained Real      | 0.61           |
> | Trained Synthetic | 0.47           |
> | Trained Augmented | **0.71** |
>
> *New segmentation results: new baseline models for the segmentation task for better comparison of effect of synthetic data on segmentation model*

---

> > ### Author Response · Authors · 2023-08-22
> >
> > 4. > Missing evaluation with DiffCollage (Table 1).
> >    >
> >
> > Thank you for suggestion this evaluaiton. We compared the Improved Precision (IP) and the Improved Recall (IR) of DiffCollage with our method and updated table 1 that you can also see below. Interestingly, our method performs significantly better wrt. IR which might be a result of the tiling artifacts produced by DiffCollage.
> >
> > |                        | **IP** ↑ | **IR** ↑ |
> > | ---------------------- | --------------- | --------------- |
> > | DiffCollage            | 0.94            | 0.22            |
> > | DiffInfinite (b)       | 0.95            | **0.48**  |
> > | DiffInfinite (c)       | **0.98**  | 0.44            |
> > | DiffInfinite (b) & (c) | **0.98**  | 0.33            |
> >
> > - DiffInfinite (a) first generates a mask and secondly an image of size $512 \times 512$ (see table below)
> > - DiffInfinite (b) uses the real masks to generate images of $2048 \times 2048$
> > - DiffInfinite (c) first generates a mask and secondly an image of size $2048 \times 2048$
> >
> > For the sake of completeness, we also updated the memorization metrics to compare different variants of DiffInfinite to DiffCollage. The updated Table 3 in Appendix C reads as follows
> >
> > |                        | **A**↑ _tiled_ | **A**↑ _resized_ | **C_T** ↑↓ _tiled_ | **C_T** ↑↓ _resized_ |
> > | ---------------------- | ----------------------- | ------------------------- | ---------------------------- | ------------------------------ |
> > | DiffCollage            | **0.89**          | 0.97                      | 11.02                        | **7.00**                 |
> > | DiffInfinite (a)       | 0.86                    | -                         | 4.99                         | -                              |
> > | DiffInfinite (b)       | 0.86                    | 0.97                      | **3.29**               | 8.11                           |
> > | DiffInfinite (c)       | 0.86                    | **0.98**            | 9.61                         | 11.56                          |
> > | DiffInfinite (b) & (c) | 0.87                    | 0.95                      | 5.31                         | 10.96                          |
> >
> > For the methods that output a large image of size $2048\times 2048$ we consider
> >
> > - *tiled* patches resulting in $16$ patches of size $512\times 512$ per large image
> > - *resized* image resulting in $200$ images of size $512\times 512$
> >
> > We are grateful for the comprehensive review and valuable suggestions. We believe our revisions fully address the raised concerns, and we hope this enhanced version of our paper will be received favorably.
> >
> > In good spirits,
> >
> > The authors

---

> > > ### Comment · Reviewer_Wamo · 2023-08-25
> > >
> > > I would like to thank the authors for taking time to adequately respond to my concerns, all of which have been addressed. The extra evaluations reveal the proposed model still achieves significant gains over a related method, better contextualizing its utility. I am happy to raise my score in this regard.

---

### Official Review · Reviewer_kGnb · 2023-07-20
**This work describes a method for generation of synthetic tissue images and segmentation masks. It proposes a diffusion model that can generate large images and evaluates the performance of the model both qualitatively and quantitatively.**

**Rating:** 7
**Confidence:** 3

**Strengths:**

-	The method is able to produce large synthetic tissue images and their segmentation masks.
-	The method includes an outpainting method that can be parallelized to speed up data generation process.
-	Evaluation with experienced pathologists as well as quantitative evaluation with fidelity and anti-copying metrics.


**Additional Feedback:**

.

**Clarity:**

The paper is well written. However, a more detailed figure describing some of the components and steps of the method would be useful, e.g., projection (proj) operations and random patch diffusion.

**Correctness:**

The claims appear to be correct. Their experimental results support the claims.

**Documentation:**

The paper is well written. The supplemental material describes the data collection process as well as provides more details about the metrics, the model training process, and sampling.

**Ethics:**

No ethical concerns.

**Limitations:**

The authors note the need for further optimizations in synthetic data generation process for some applications (like training deep learning models).

**Opportunities For Improvement:**

A key use of synthetic data is to augment training data for deep learning methods, especially when real training data is small. The experimental evaluation of this use is quite limited. The authors train a model for four-class segmentation and present an F1 score. Without a baseline comparison, it is difficult to assess if the F1 score is good and whether the synthetic data improves the performance of the deep learning model. The authors should create a training dataset from a subset of the real histological dataset (Table 3 in the supplement document) as training data and train a state-of-the-art segmentation model without the synthetic dataset as well as with the synthetic dataset.

**Relation To Prior Work:**

The related work section covers prior work in several related areas.

**Summary And Contributions:**

This work describes a method for generation of synthetic tissue images and segmentation masks. It proposes a diffusion model that can generate large images, unlike previous works that limited to small image patches.  An experimental evaluation of the method shows it performs well with respect to several fidelity metrics and data memorization metrics. The method is also evaluated qualitatively by expert pathologists.

---

> ### Author Response · Authors · 2023-08-22
>
> Dear Reviewer kGNb,
>
> We thank you for their time and valuable feedback. We try to address all your comments in the responsed below.
>
> 1. > Difficult interpretation of the downstream task without baseline
>    >
>
> We concur with the reviewer’s feedback and have integrated the recommended experiments. Accordingly, we trained a model using the same real data patch dataset which was also employed for training the diffusion model. To assess the model’s ability to generalize to a distinct cohort of lung patients from a different hospital, we utilized an internal dataset of 13 additional whole slide images. This allowed us to comprehensively evaluate and compare performance between models trained on both real data and on synthetic data, as outlined in the updated Section 5.3. While it remains true that the performance of the model exclusively trained on synthetic data continues to lag behind the model trained exclusively on real data, we present evidence that supplementing the real data with synthetic data enhances the model's capacity to generalize to the external dataset from a different distribution.
>
> |                   | **IH2**  |
> | ----------------- | -------------- |
> | Trained Real      | 0.61           |
> | Trained Synthetic | 0.47           |
> | Trained Augmented | **0.71** |
>
> *New segmentation results: new baseline models for the segmentation task for better comparison of effect of synthetic data on segmentation model.*
>
> 2. > More detailed description of model components
>    >
>
> We updated the flow of Figure 2 along with a more detailed description of the time tracking below "Random path diffusion" in Section 4.3 and Algorithm 2. Moreover, we defined the three variants of our methods evaluated in Table 1:
>
> - DiffInfinite (a) first generates a mask and secondly an image of size $512 \times 512$
> - DiffInfinite (b) uses the real masks to generate images of $2048 \times 2048$
> - DiffInfinite (c) first generates a mask and secondly an image of size $2048 \times 2048$
>
> We are grateful for the comprehensive review and valuable suggestions. We believe our revisions fully address the raised concerns, and we hope this enhanced version of our paper will be received favorably.
>
> In good spirits,
>
> The authors

---

> > ### Comment · Reviewer_LR7b · 2023-08-30
> >
> > I thank the authors for addressing my concerns and have raised my score accordingly.

---

### Official Review · Reviewer_LR7b · 2023-07-26
**DiffInfinite**

**Rating:** 9
**Confidence:** 3

**Strengths:**

The ability to generate such images to the degree they can reliably fool experts is very impressive, especially across scales. A lot of work has been done to characterize the images, and given they come from medical data, I do especially appreciate the attention to dangers of memorization.

**Additional Feedback:**

Nice work!

**Clarity:**

Generally yes, but the paper does assume a high level of familiarity specifically with diffusion models as described above

**Correctness:**

To the degree I can tell without data or code access, I have no reason to believe not.

**Documentation:**

Code has not yet been provided so this is not yet possible to assess

**Ethics:**

None beyond the potentfial for misuse described above

**Limitations:**

In an era of scientific reproducibility crisis where image faking is a common concern, I worry about the potential of misuse of this technology to create images that (wrongly) support particular biological and especially medical findings - it would be nice to have the authors discuss this.

**Opportunities For Improvement:**

- As someone not deeply an expert on diffusion models, I found the actual model architecture description confusing, since in many cases it just cites out to other papers rather than deeply describing. While it need not be beginner-level, a higher level overview and a more thorough diagram describing order of training and data flow (based on Table 1, multiple flows are possible) through the various parts of the overall architecture seems needed to me, and beneficial even if not strictly needed by people who spend more time thinking about diffusion models than I personally do.
- Neither code nor data is yet provided, so it isn't clear to me how this paper, while impressive, is a natural fit for this track specifically; I suppose it falls under "Data generators" but it would be nice to add a clearer sentence or two to why it's in this session.
-- One of the major things I think IS potentially a benchmark here is the performance of their synthetic data in a downstream training task (5.3), but this section is under-explained in terms of the actual task being performed and how it's being evaluated, and it's unclear how good a score 64 is for this particular task. Could the same task be done with real data (presumably smaller in size; could do a smaller set of DiffInfinite training to apples-to-apples the comparison) and relative performance evaluated?

**Relation To Prior Work:**

Yes

**Summary And Contributions:**

In this paper, the authors lay out a new system for generating large scale whole slide pathology images and doing better image prediction across large scale images. Such images are critical for medical tasks, and the ability to generate realistic images would be helpful for many tasks (though dangerous in others).

---

> ### Author Response · Authors · 2023-08-22
>
> Dear Reviewer LR7b,
>
> thank you for your insightful comments and valuable feedback. See below for detailed answers to your questions and concerns.
>
> 1. > Shortcomings of model description
>    >
>
> We updated the flow of Figure 2 along with a more detailed description of the time tracking in Section 4.3 and Algorithm 2.
>
> 2. > Based on Table 1, multiple flows are possible
>    >
>
> We defined the three variants of our methods evaluated in Table 1:
>
> - DiffInfinite (a) first generates a mask and secondly an image of size $512 \times 512$
> - DiffInfinite (b) uses the real masks to generate images of $2048 \times 2048$
> - DiffInfinite (c) first generates a mask and secondly an image of size $2048 \times 2048$
>
> 3. > Neither code nor data is yet provided
>    >
>
> The executable code is now provided with the updated manuscript. In addition, we provide an interactive notebook to test and run the model as well as a set of data samples. All artifacts can be anonymously accessed at
>
> - Algorithm code: [https://github.com/diffinfinite/diffinfinite](https://github.com/diffinfinite/diffinfinite)
> - Model weights: [https://github.com/diffinfinite/diffinfinite/blob/master/download.py](https://github.com/diffinfinite/diffinfinite/blob/master/download.py) (can be downloaded via script)
> - Set of example images: [https://github.com/diffinfinite/diffinfinite/tree/master/results](https://github.com/diffinfinite/diffinfinite/tree/master/results)
> - Interactive notebook to interact with the model: [https://colab.research.google.com/github/diffinfinite/diffinfinite/blob/master/main.ipynb](https://colab.research.google.com/github/diffinfinite/diffinfinite/blob/master/main.ipynb)
>
> 4. > It isn't clear to me how this paper, while impressive, is a natural fit for this track
>    >
>
> First of all, we are very happy that you think our work is impressive. We’d like to highlight that the call for papers explicitly includes ”data generators” and we  believe it is relevant to the Datasets & Benchmarks track as we identify and make contributions  towards addressing data-centric issues specific to histopatological imaging, namely: reducing  the need for costly manual annotation of whole slide images; minimising the usage of real  patient data in downstream machine learning tasks; and providing the ability to efficiently  generate synthetic images at resolution which matches real WSI much more closely than before.  Indeed, several papers addressing wider data-centric issues have been accepted in this track in previous years, including  https://openreview.net/forum?id=v3yM5zVzP4C, which similarly addresses a data-centric problem and then proposes an improved way of curating data, and  https://openreview.net/forum?id=zNQBIBKJRkd  which discusses trends in dataset and benchmark curation within the machine learning community and which received the best paper award in 2021. While we provide some synthetic data samples to  illustrate the quality of the generated data, our contribution goes beyond providing synthetic dataset. Our method and codebase, which we make publicly available, allows the community to efficiently generate any number of unique datapoints, while providing flexibility in terms of target resolution and spatial distribution, while at the same time, overcoming the data issues outlined above.
>
> 5. > One of the major things I think IS potentially a benchmark here is the performance of their synthetic data in a downstream training task
>    >
>
> Even though, this potentially is a benchmark, our intention here is different, as we provide a downstream task in order to add another dimension to the evaluation of the generated data. We evaluate if the generated data is able to improve the performance of a downstream task, thereby indicating the generation of additional information that a downstream model can pick up.

---

> > ### Author Response · Authors · 2023-08-22
> >
> > 6. > Section 5.3 is under-explained in terms of the actual task being performed and how it's being evaluated
> >    > and it's unclear how good a score 64 is for this particular task
> >    >
> >
> > Addressing your criticism of section 5.3 we rewrote this section and carried out additional experiments.
> > We retrained a model using the same real data patch dataset which was also employed for training the diffusion model. To assess the model’s ability to generalize to a distinct cohort of lung patients from a different hospital, we utilized an internal dataset of 13 additional whole slide images. This allowed us to comprehensively evaluate and *compare performance* between models trained on both real data and on synthetic data, as outlined in the updated Section 5.3. While it remains true that the performance of the model exclusively trained on synthetic data continues to lag behind the model trained exclusively on real data, we present evidence that supplementing the real data with synthetic data enhances the model's capacity to generalize to the external dataset from a different distribution.
> >
> > |                   | **IH2**  |
> > | ----------------- | -------------- |
> > | Trained Real      | 0.61           |
> > | Trained Synthetic | 0.47           |
> > | Trained Augmented | **0.71** |
> >
> > *New segmentation results: new baseline models for the segmentation task for better comparison of effect of synthetic data on segmentation model.*
> >
> > 7. > I worry about the potential of misuse of this technology to create images that (wrongly) support particular biological and especially medical findings
> >    >
> >
> > Thank you for pointing out these serious concerns. In our opinion, having the generated data evaluated by experienced pathologists is the most convincing way to ensure that the model generates to a large extent biologically plausible data. In addition, extensive evaluation on real data of any machine learning model (partly) trained on generated data can prevent harm, but we also want to point out that there is no general guideline yet to prevent the misuse of generative models.
> >
> > We are grateful for the comprehensive review and valuable suggestions. We believe our revisions fully address the raised concerns, and we hope this enhanced version of our paper will be received favorably.
> >
> > In good spirits,
> >
> > The authors

---

### Author Response · Authors · 2023-08-22

Dear Reviewers,

We extend our gratitude for the comprehensive reviews and constructive feedback. Your insights have been pivotal in refining our manuscript. Here, we briefly summarize the strengths acknowledged across reviews and address the prevalent areas of improvement.

### Consensus on Strengths:

**Significance of Synthetic WSIs:** The overarching consensus underscores the value of our synthetic histological images, given the inherent challenges and costs associated with obtaining real Whole Slide Images.

**Innovative Sampling and Generation Techniques:** Our novel sampling methodology, which circumvents issues observed in previous methods through greater flexibility, received collective appreciation. Additionally, the quality, scale, and diversity of images generated by our model were acknowledged, notably without the pitfalls of memorization, despite using sensitive patient data.

**Rigorous Evaluation:** A recurring highlight was the thoroughness of our evaluation, both quantitatively and qualitatively. The involvement of experienced pathologists in the validation process, combined with various fidelity and anti-copying metrics, was highlighted as a particular strength of this study.

**Contribution to the Field:** Our work's unique ability to generate high-resolution images in the complex domain of histopathology, especially when compared to existing methods, was recognized as a significant advancement.

### Addressing Key Concerns:

**Relevance to the Datset and Benchmark Track:** Some reviewers were uncertain about the relevance to D\&B track. We like to highlight that on the one hand our released model fits well with intended topic of novel data generators and is well in line with submissions to previous years (we provide a sample list in the reviewer response). On the other hand we want to add that the trained model can be used by interested research groups to generate synthetic datasets for data augmentation in different downstream tasks.

**Clarity on Downstream Task:** We acknowledge that the previous description of the downstream task fell short. Responding to the feedback, we've enriched our manuscript with a more detailed description and incorporated additional experiments for both segmentation and a novel classification task.

**Public Model Code and Synthetic Data Release:** Addressing the call for transparency, we have released

- Algorithm code: [https://github.com/diffinfinite/diffinfinite](https://github.com/diffinfinite/diffinfinite)
- Model weights: [https://github.com/diffinfinite/diffinfinite/blob/master/download.py](https://github.com/diffinfinite/diffinfinite/blob/master/download.py) (can be downloaded via script)
- Set of example images: [https://github.com/diffinfinite/diffinfinite/tree/master/results](https://github.com/diffinfinite/diffinfinite/tree/master/results)
- Interactive notebook to interact with the model: [https://colab.research.google.com/github/diffinfinite/diffinfinite/blob/master/main.ipynb](https://colab.research.google.com/github/diffinfinite/diffinfinite/blob/master/main.ipynb)

to verify the quality of the generated samples.

In closing, we genuinely value the collective insights shared, which we have used to hone our work further. We trust that our revised manuscript and this response resonate with the vision of NeurIPS and meet your esteemed standards.

In good spirits,

The authors

---

### Decision · Program_Chairs · 2023-09-22

**Decision:**

Accept (Spotlight)

**Comment:**

Strong contribution, and a well-written paper. It received all positive reviews. The paper is recommended for acceptance.